# Nanoconfinement enabled non-covalently decorated MXene membranes for ion-sieving

Yuan Kang[1], Ting Hu[1], Yuqi Wang [2], Kaiqiang He[1], Zhuyuan Wang[3], Yvonne Hora[1], Wang Zhao[1], Rongming Xu[4], Yu Chen [5], Zongli Xie [6], Huanting Wang [1], Qinfen Gu [7] ✉ & Xiwang Zhang [1,3] ✉

Covalent modification is commonly used to tune the channel size and functionality of 2D membranes. However, common synthesis strategies used to produce such modifications are known to disrupt the structure of the membranes. Herein, we report less intrusive yet equally effective non-covalent modifications on $Ti_3C_2T_x$ MXene membranes by a solvent treatment, where the channels are robustly decorated by protic solvents via hydrogen bond network. The densely functionalized (-O, -F, -OH) $Ti_3C_2T_x$ channel allows multiple hydrogen bond establishment and its sub-1-nm size induces a nanoconfinement effect to greatly strengthen these interactions by maintaining solvent-MXene distance and solvent orientation. In sub-1-nm ion sieving and separation, as-decorated membranes exhibit stable ion rejection, and proton-cation ($H^+/M^{n+}$) selectivity that is up to 50 times and 30 times, respectively, higher than that of pristine membranes. It demonstrates the feasibility of non-covalent methods as a broad modification alternative for nanochannels integrated in energy-, resource- and environment-related applications.

Membranes based on rolled, perforated, or stacked 2D materials possess channels in 1-nm and sub-1-nm scales and present relevance and significance across disciplines[1–3]. With channel dimension approaching the size of most hydrated ions (<1 nm), gases (<0.5 nm), and solvent molecules (<2 nm), 2D membranes allow controllable mass separation, a core step to drive critical technologies in resource recovery, environmental remediation, energy conversion, and storage. Particularly in the stack form, the membranes have regularly aligned neighboring layers so that interlayer channels with narrow size distribution can be formed to induce high selectivity between physiochemically similar species. Accompanying the ongoing material exploration on graphene derivatives, metal nitrides/carbides (MXene), metal-organic frameworks (MOFs), covalent organic frameworks (COF), a few such channels have been composed with primary width between 0.3 and 2 nm, and functioned well in several separation processes[4–7].

To extend their use for numerous other separating purposes and in various media (e.g., dry, wet, or solvated), these pristine channels always need customized tailoring on their size and/or surface chemistry to satisfy each specific requirement. A typical example is aquatic-used graphene oxide (GO) and MXene channels that show effective widths of ~0.6 to 0.8 nm[1,6]. While this intrinsic size meshes out polyatomic ions and organic pollutants over 1 nm in wastewater purification, it needs an additional reduction to below 0.6 nm to enable monoatomic ion sieving as in desalination, and proton isolation as in redox batteries[3]. The majority of such modifications are achieved via covalent crosslinking or decoration, where guest species are chemically bonded to both or one side of the channels so that desired channel size and functionalities can be obtained[1,4]. While providing reliable modification efficacy, covalent methods are discouraged from fulfilling their decoration potential largely due to the emergence of

[1]Department of Chemical and Biological Engineering, Monash University, Clayton, VIC 3800, Australia. [2]School of Materials Science and Engineering, Zhejiang University, 310058 Zhejiang, China. [3]UQ Dow Centre for Sustainable Engineering Innovation, School of Chemical Engineering, The University of Queensland, St. Lucia, QLD 4072, Australia. [4]School of the Environment, Nanjing University, 210023 Nanjing, China. [5]Monash Centre for Electron Microscopy, Monash University, Clayton, VIC 3800, Australia. [6]CSIRO Manufacturing, Private Bag 10, Clayton South 3169, Australia. [7]Australian Synchrotron, ANSTO, Clayton, VIC 3168, Australia. ✉e-mail: qinfeng@ansto.gov.au; xiwang.zhang@uq.edu.au

performance-compromising non-idealities to membrane structure in the reaction process. These include unwanted defects, corrugations, and interrupted channel alignment caused by harsh reaction conditions (e.g., high temperature and strong acid/base) and volatile by-products[8–10].

Equally interesting, but much less explored, is to modify nanochannels in non-covalent routes. Known as a collective of van der Waals forces, electrostatic interactions, hydrogen bonds, and π–π interaction, non-covalent interactions universally form within a 2 nm threshold intermolecular distance, promising easy membrane modifications without major structural compromise. Nevertheless, their low strength (−0.4 eV) and exponential weakening beyond the threshold distance present a potential instability problem for non-covalent modifications, making them less a favor than covalent ones (−1.0 eV) in wide conventional channels. Recent research on nanofluid, however, proves otherwise by discovering a nanoconfinement effect where non-covalent affinity can be boosted by one order of magnitude in constrained geometry[11]. With most 2D membranes possessing 1-nm channels, such a phenomenon can be utilized to enable non-covalent decorations.

Considering its stable 1-nm channel size[12], we herein take 2D $Ti_3C_2T_x$ ($T_x$ represents −O, −OH, and −F) MXene as a platform and propose an almost non-intrusive, effective non-covalent decoration strategy by facile solvent treatment. When streaming into MXene channels, organic molecules, particularly protic ethanol, can attach themselves onto the functional groups of MXene surface via hydrogen bond (H-bond) network. Owing to the sub-1-nm space and the establishment of multiple H-bonds, the confined ethanol is able to set up closer contact with MXene channels than in bulk solution to generate a much stronger and more stable interaction (−1.29 eV). The as-decorated nanochannels exhibit improved ion sieving ability, proton-ion selectivity, and long-term stability, providing a new, applicable category of modification methods for 1-nm and sub-1-nm channels.

## Results and discussion

### MXene membrane preparation and characterizations

Highly crystallized, monolayered $Ti_3C_2T_x$ nanosheets (Supplementary Fig. 1) with low oxidation degree and small lateral size distribution (Supplementary Fig. 2) were etched from parent MAX phase $Ti_3AlC_2$[13,14]. When self-assembled into laminar membranes, the rearranged nanosheets illustrated a defect-free surface and layer-like cross-section structure, indicating the formation of internal 2D interlayer channels (Supplementary Fig. 3). By a simple solvent immersion method, these untreated MXene membranes (Untreated-M) turned into ethanol-treated membranes (EtOH-M, Supplementary Fig. 4). In accordance with previous studies, X-ray photoelectron spectroscopy (XPS) and energy-dispersive X-ray (EDX) analysis revealed the membrane chemical composition, comprising −O, −OH and −F groups linked to Ti and C atoms across the basal plane and around the edge of nanosheets (Fig. 1a, b, and Supplementary Fig. 5)[15,16]. These characteristic groups were also detected in Fourier-transform infrared spectroscopy (FTIR), but with a clear red shift at some peaks after ethanol treatment (Fig. 1c). Two major peaks assigned to −OH (3460 cm$^{-1}$), and Ti−O (626 cm$^{-1}$) moved to lower wavenumber of 3410 and 606 cm$^{-1}$[17–19]. This was largely because of the hydrogen bond formed between them and the −OH from the inserted ethanol, which causes the electron cloud density change and elongation of involving bonds. Consequently, the strength of the bonds will be decreased to show a lower stretching vibration frequency in FTIR[20–22]. Also, two joint peaks emerging at 2920 and 2850 cm$^{-1}$, which are normally attributed to −CH$_3$/−CH$_2$−, also prove the successful insertion of ethanol molecules into MXene membranes.

Further X-ray diffraction (XRD, Fig. 1d and Supplementary Fig. 6) characterization on the membrane showed two-fold implications on its channel interiors. The hydrated interlayer channel size remained the same at around 16.5 Å regardless of the ethanol treatment, suggesting likely modifications via decoration (MX-EtOH) rather than crosslinking

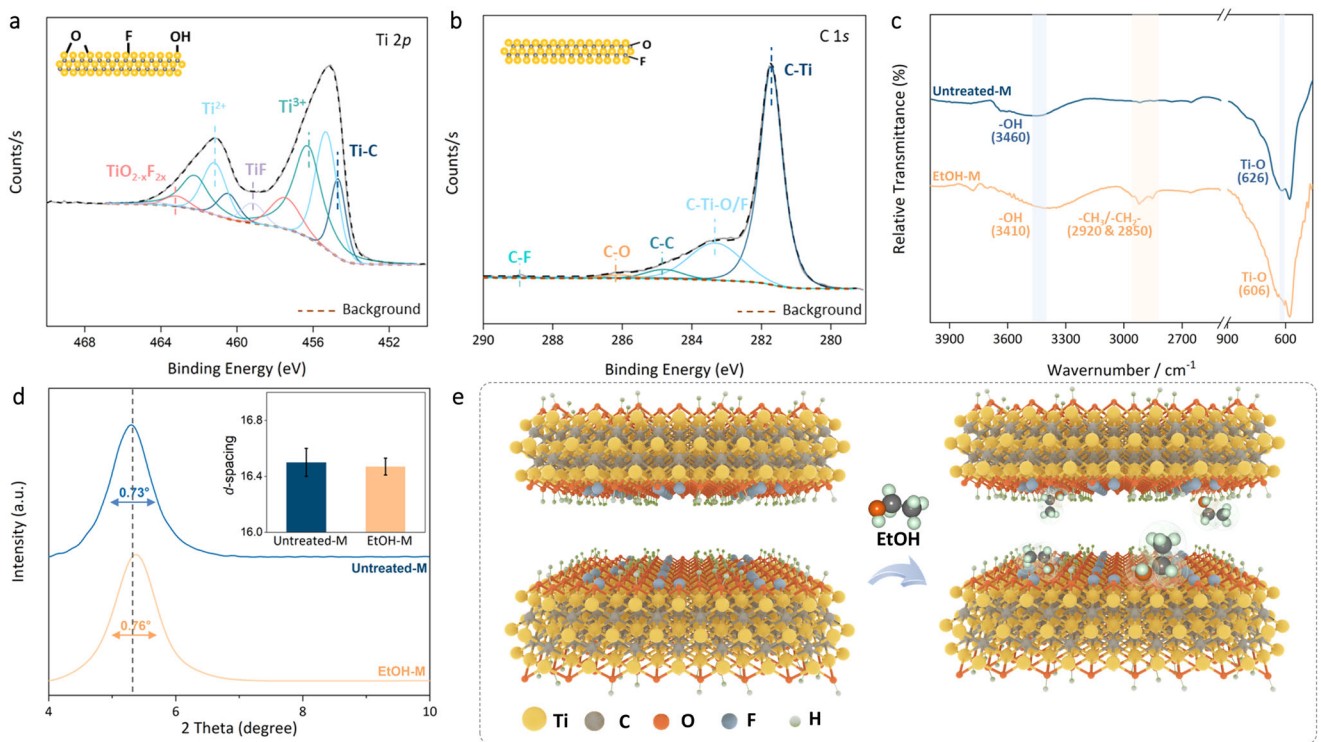

**Fig. 1 | Physicochemical characterizations on $Ti_3C_2T_x$ MXene membranes.** **a**, **b** XPS elemental analysis of Ti (Ti 2$p$) and C (C 1$s$) in MXene. **c** FTIR spectra of Untreated-M and EtOH-M. The highlighted regions indicate FTIR peak changes after EtOH treatment. **d** XRD patterns of Untreated-M and EtOH-M. Inset, corresponding $d$-spacing is calculated from the patterns. **e** Schematic of ethanol decoration in $Ti_3C_2T_x$ channels. The error bars in this figure represent the standard deviations of three parallel tests.

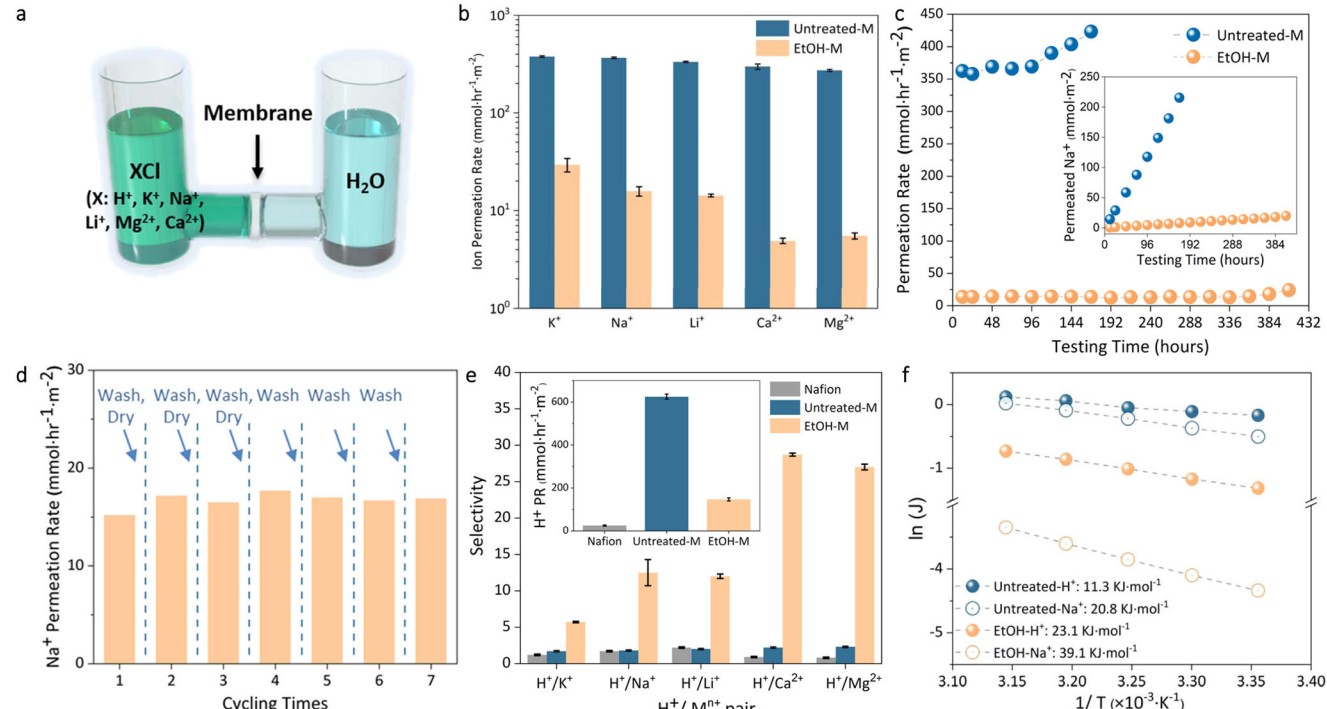

**Fig. 2 | Ion sieving performance of Untreated-M and EtOH-M. a** Ion permeation through MXene membranes in a U-shape device. **b** The permeation rate of $K^+$, $Na^+$, $Li^+$, $Ca^{2+}$ and $Mg^{2+}$ (0.2 M) through Untreated-M and EtOH-M. **c** Long-term $Na^+$ permeation rate through Untreated-M and EtOH-M. Inset, accumulated permeated $Na^+$ along with testing time. **d** $Na^+$ permeation rate in cyclic tests. **e** $H^+/M^{n+}$ ($K^+$, $Na^+$, $Li^+$, $Ca^{2+}$, and $Mg^{2+}$) selectivity in Untreated-M, EtOH-M, and a commercial Nafion membrane. Inset, $H^+$ permeation rate in these membranes. **f** Arrhenius plots of temperature against $H^+$ and $Na^+$ permeation rate for Untreated-M and EtOH-M. The error bars in this figure represent the standard deviations of three parallel tests.

(MX-EtOH-MX), because the latter usually results in smaller interlayer distances by XRD[23]. More importantly, an almost unchanged peak full width at half maximum (FWHM) was recorded before (0.73°) and after (0.76°) solvent insertion[24]. It proves that unlike in most covalent cases, non-covalent membrane modifications can be achieved without disrupting the channel order, as depicted in Fig. 1e.

## Ion sieving and separating performance of EtOH-M

The ion sieving performance of both untreated and EtOH-M was tested and compared towards a few common ions to evaluate the decoration efficiency (Fig. 2a). For a 300-nm-thick untreated membrane, the permeation rate of $K^+$, $Na^+$, $Li^+$, $Ca^{2+}$, and $Mg^{2+}$ reached 377.5, 367.5, 333.7, 298.3 and 272.5 mmol $h^{-1}$ $m^{-2}$, respectively, similar to previous studies (Fig. 2b)[25,26]. The relatively fast ion transport is reasonable considering the effective channel size of 7.7 Å (16.5−8.8 Å, the thickness of a single $Ti_3C_2T_x$ nanosheet)[12]. Once decorated with ethanol, it is expected that part of the channel is occupied to narrow down the effective channel width and obstruct the passage of ions. Accordingly, membrane ion rejection improved by 15 to 60 times, allowing only 25.4, 15.8, 14.3, 4.9, and 5.5 mmol $h^{-1}$ $m^{-2}$ rate for corresponding ions. Such improved ion sieving performance is comparable to that of previously reported covalently and physically modified 2D membrane channels, suggesting similar decoration efficacy via non-covalent methods (Supplemental Table 1). Considering that non-covalent interactions including hydrogen bonding are generally low in strength, long-term and cyclic tests were then conducted using $Na^+$ solution to examine the stability of the channel decoration strategy. In a 2-week-long continuous test, the permeation rate of $Na^+$ in Untreated-M showed an obvious upward trend after the 4th day, implying the gradual $Ti_3C_2T_x$ oxidation that compromised its ion-rejecting ability[25]. In opposition, the rate remained rather stable for EtOH-M at around 15.0 mmol $h^{-1}$ $m^{-2}$ until the 17th day (Fig. 2c). Such

contrast not only proved the sufficient affinity of decorated ethanol to the channel but also highlighted its bonus role to prevent $Ti_3C_2T_x$ membranes from degrading via a "molecular shielding" effect (Supplementary Fig. 7)[27]. In cyclic experiments using $Na^+$ solution, EtOH-M went through washing and drying (or drying only) after each testing session, to mimic the real membrane operating situations (Fig. 2d). Despite a minor rise between the 1st and the 2nd cycle, the permeation rate of $Na^+$ is stable during operation mode. It is also worth mentioning that to increase solvent treatment time and membrane thickness had limited influence on membrane ion sieving performance, implying that the intake of ethanol into the channel would soon reach a capacity (Supplementary Fig. 8). This is understandable considering the stiff configuration of ethanol that only allows limited rotation and vibration. After a few molecules are anchored, the diffusion of the following ethanol will be impeded as they cannot shrink or squash like hydrated ions to pass the already obstructed channel[26,28].

Besides, to demonstrate the decoration effect on channel selectivity, we then tested membranes' ability to separate protons ($H^+$, 2.8 Å) from the above salts (>6.6 Å, $K^+$) before and after solvent treatment. Figure 2e shows that proton transport was also retarded in the EtOH-anchored MXene channels, albeit by a much smaller extent compared to other salt transport. While the permeation rate of the proton was reduced from 625.4 to 146.9 mmol $h^{-1}$ $m^{-2}$ in EtOH-M, its selectivity against $K^+$, $Na^+$, $Li^+$, $Ca^{2+}$, and $Mg^{2+}$ increased from around 2.0−5.7, 12.5, 12.0, 28.7, 27.0 respectively. The enhanced selectivity was rationalized by the Arrhenius plot of temperature against ion permeation rate (Fig. 2f). Although both ions experienced higher energy barrier in EtOH-M than in Untreated-M, the activation energy increase for $Na^+$ (from 20.8 to 39.1 kJ $mol^{-1}$) was obviously more pronounced than that for $H^+$ (11.3−23.1 kJ $mol^{-1}$). This implies that the decorated EtOH molecules created a substantially larger barrier for $Na^+$ to dehydrate while they only moderately interfered with the "hopping" (Grotthuss-like

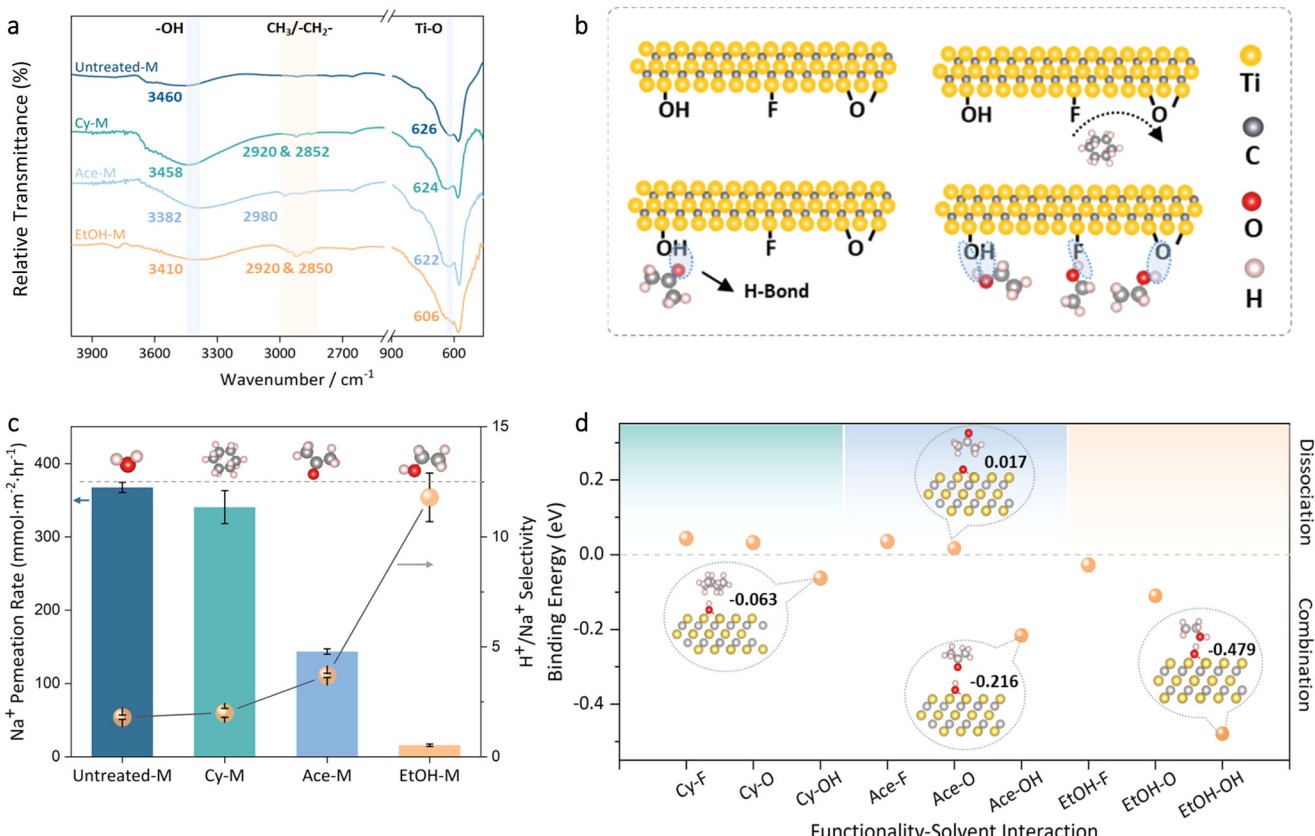

**Fig. 3 | Solvent-dependent non-covalent channel decoration efficacy and ion sieving performance. a** FTIR spectra of Untreated-M, Cy-M, Ace-M, and EtOH-M. The highlighted regions indicate FTIR peak changes after solvent treatment. **b** Schematic of the possible hydrogen bonding between MXene nanosheets and cyclohexane, acetone, and ethanol. **c** Na⁺ sieving performance of various MXene membranes. **d** DFT calculation of binding energy between MXene functionalities (−O, −F, −OH) and different solvents on "1-on-1" mode. The error bars in this figure represent the standard deviations of three parallel tests.

transport) of H⁺. Such high proton permeance and proton-cation selectivity of EtOH-M far exceed those of current Nafion membranes (proton permeation rate: 25.1 mmol h⁻¹ m⁻²; Selectivity: <2.2), and show its potentials as proton-selective membranes for electrolysis and flow batteries.

**Solvent-dependent non-covalent decoration efficacy**

To further verify the non-covalent (H-bond) nature of the proposed solvent decoration strategy, we then prepared and compared Ti₃C₂Tₓ membranes treated by non-polar cyclohexane (Cy-M), and polar yet aprotic acetone (Ace-M) besides aprotic EtOH-M. While having a similar size to ethanol (4.4 Å), cyclohexane (4.7 Å) and acetone (4.6 Å) are chemically different, which supposedly leads to varying inherent capabilities of establishing an H-bond network. This hypothesis was first supported by FTIR results in Fig. 3a. Unlike EtOH-M which showed multiple peak redshifts, Cy-M displayed an almost identical spectrum to that of Untreated-M, and Ace-M spectrum only recorded a −OH shift from 3460 to 3382 cm⁻¹. Meanwhile, while ethanol generated two obvious characteristic C−H (−CH₃ or −CH₂−) peaks between 3000 and 2800 cm⁻¹, acetone and cyclohexane could only generate much weaker ones, implying their lower or even no adsorption into MXene channels. This difference can be well explained by the H-bond forming mechanism: hydrogen (H) directly connected to a highly electronegative atom (X, donor, usually N, O, F, and S) is approached by another alike atom with lone-pair electrons (Y, acceptor)[29]. As a non-polar hydrocarbon solvent, Cyclohexane is inept at H-bond formation (Fig. 3b). With a polar carbonyl yet no active proton, acetone can barely form H-bond with −OH on MXene (Ace = O ⋯ H−O−MX). As for ethanol, its hydroxyl enables its versatile binding with MXene via −O

(Et−O−H ⋯ O−MX), −F (Et−O−H ⋯ F−MX), and especially −OH (Et−O−H ⋯ OH−MX and Et−HO ⋯ H−O−MX). Higher H-bond forming probability will translate into larger decoration coverage in MXene channels, and ultimately superior ion-sieving and proton-separating performance (Supplementary Fig. 9). Correspondingly, EtOH-M was found to have a 23-time Na⁺ rejection enhancement relative to Untreated-M, but Cy-M and Ace-M saw only 1.2-times and minor 2.6-times increase, respectively (Fig. 3c). At the same time, EtOH-M and Ace-M showed an H⁺/Na⁺ selectivity of around 12 and 4, respectively, while Untreated-M and Cy-M could only separate H⁺ and Na⁺ by a factor of 2.

To quantitatively analyze the decoration difference, we then employed density-functional theory (DFT) to study each solvent-functionality pair. This was achieved by calculating the binding energy of each pair in a simplified "1-on-1" mode where a single solvent molecule is allowed to stabilize on one MXene functional group for all three Cy-M, Ace-M, and EtOH-M samples (Fig. 3d and Supplementary Table 2). In good agreement with experimental findings, the DFT results revealed two obvious characteristics. (1) all the pairs involving ethanol demonstrated negative binding energy ($E_{abs}$ < 0), which again proves the tendency of ethanol to be adsorbed onto MXene via various surface groups. Comparatively, acetone and cyclohexane could only set up this spontaneous link via Ace-OH (−0.216 eV) and Cy-OH (−0.063 eV) pairs, implying their lower chance to decorate the channel; (2) When paired with any surface groups (−F, −O and −OH) from MXene nanosheets, ethanol always generated stronger combination than the other two. Especially in hydroxyl-containing pairs, EtOH-OH showed the highest $E_{abs}$ of −0.479 eV, being 2 and 7 times as much as that of Ace-OH and Cy-OH, respectively. In addition, the dynamic and thermodynamic stabilities of the EtOH decorated MXene structure are

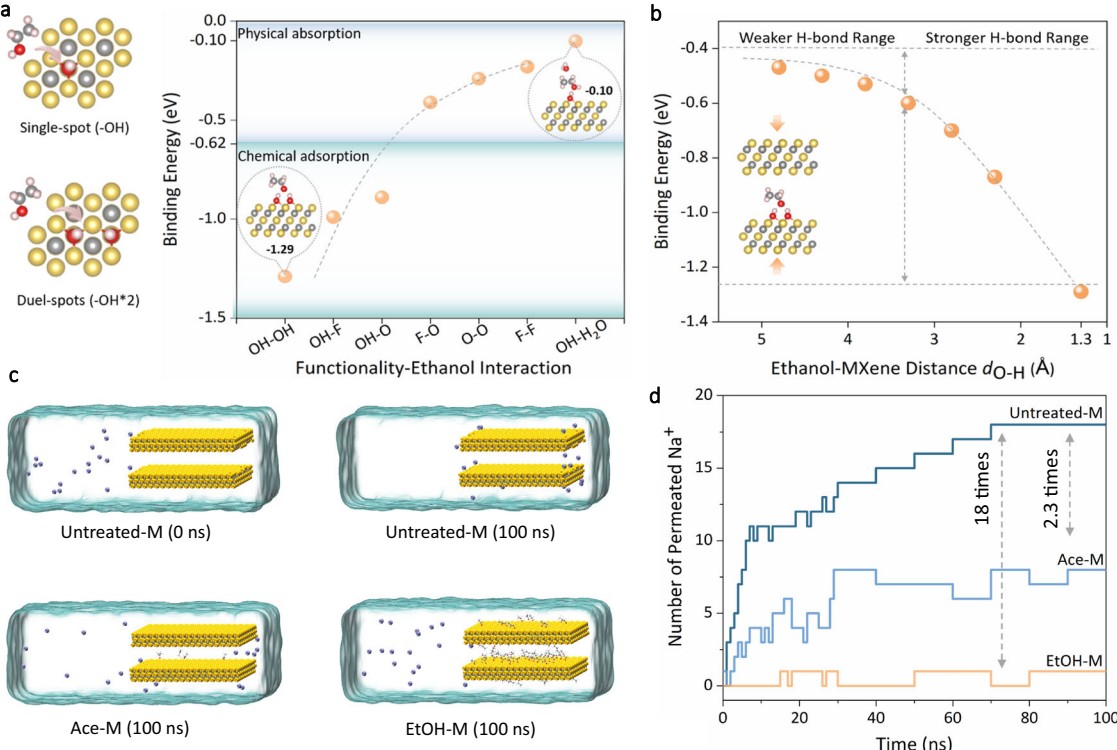

**Fig. 4 | DFT and MD simulation on channel stability and ion sieving ability.**
**a** DFT calculation of binding energy between ethanol and MXene functional groups in "one-between-two" mode. **b** The effect of ethanol–MXene distance on its binding strength. **c** Schematic and **d** results of Na$^+$ permeation through the simulated channel of Untreated-M, Ace-M, and EtOH-M over the course of 100 ns.

further verified by ab-initio molecular dynamics (AIMD) calculations. After a dynamic simulation of 10 ps with a time step of 1 fs, the EtOH-M structure has neither significant deformation nor EtOH molecule disassociation, which reflects its thermodynamic stability. Such higher solvent-dependent combination chance and strength not only elucidates the non-covalent origin of our decoration method but also advises that protic functional groups like −OH, whether from guest materials or host channels, is critical to non-covalently decorating MXene nanochannels in a dense and intense fashion. In addition, we also treated Ti$_3$C$_2$T$_x$ membranes with a few more solvents including methanol (MeOH), Acetaldehyde (MeCHO), and n-Hexane (Hex), which respectively, resembled EtOH, Ace, and Cy in terms of chemical properties. Similar ion-rejecting and selective performance trends were obtained, which demonstrated the reproducibility and universality of our method (Supplemental Fig. 10).

**Nanoconfinement-enabled stability and ion-sieving ability**
Although the "1-on-1" DFT calculations corroborate that appropriate solvents can absorb onto MXene channels via H-bond, the durability of these absorption-based decorations remain a major concern. This is because even for the strongest EtOH-OH, its binding strength of −0.479 eV is still below the chemical absorption range between −0.622 and −1.036 eV (60–100 kJ mol$^{-1}$), a bar generally acknowledged for stable and irreversible adsorption. To rationalize the long-term experimental stability observed for EtOH-M, we first reviewed the real-world surface chemistry of MXene nanosheets. Unlike GO and MoS$_2$ nanosheets bearing sparse functional groups, MXene surface is almost fully functionalized due to its unique progressive etching preparation method[30–32]. The massive amount of surface groups means that any solvent near MXene is highly likely to interact with more than one surface group. Therefore, we further calculated the binding energy of ethanol with two adjacent groups in an adjusted "1-between-2" mode (Fig. 4a). Apparently, the transition of any "1-on-1"

combination to corresponding "1-between-2" configuration would increase binding strength by 2–35 times (Supplementary Table 3). In particular, the incorporation of an extra group into the EtOH−OH could lead to chemically stable HO−EtOH−OH (−1.290 eV), F−EtOH−OH (−0.990 eV) and O−EtOH−OH (−0.890 eV) configurations, which was estimated to decorate around 27% of an MXene nanochannel (Supplementary Fig. 11). However, further enlarging the configuration into a "1-between-3" mode oppositely yielded decreased solvent-MXene binding energy. We attributed the looser combination to the deviating ethanol position from all three −OH in the "1-between-3" system, in which the prolonged intermolecular distance greatly weakened the overall hydrogen bond network (Supplementary Fig. 12).

Considering the high dependence of the H-bond on distance, we then investigated the effect of nanoconfinement on its formation and continuation by adjusting the ethanol−MXene distance ($d_{O-H}$) in DFT calculations. The results found ethanol most energetically stable when 1.3 Å away from −OH on MXene, and that the binding energy expectedly decreased with increased $d_{O-H}$ (Fig. 4b). More importantly, the $d_{O-H}$-eV plot displayed two different changing trends in the adjusted distance from 1.3 to 4.8 Å. Below 3.3 Å, $E_{ads}$ changed rapidly when ethanol approached or deviated from the MXene surface, showing the stronger attraction imposed by H-bond within this range. Beyond 3.3 Å, however, $E_{abs}$ would only change slightly in response to altered $d_{O-H}$. The cut-off point of 3.3 Å well reflects the short-range (<3.5 Å) feature of the H-bond and thus implies the critical role of nanoconfined space to intensify otherwise much weaker non-covalent interactions[23]. To better mimic the real-world aquatic operating situation, we also studied the interplay among MXene, ethanol, and water in the confined channel. It was revealed that the binding energy between water and the anchored ethanol molecule was only −0.1 eV, substantially lower than that of MXene−ethanol pairs (Fig. 4a). The weaker water−ethanol interaction can be attributed to the relatively fixed orientation of ethanol due to the nanoconfinement in the MXene

channels. While its −OH tends to face toward the MXene surface to maintain an energetically favorable position, this position prevents full contact between water and the polar −OH of ethanol. Therefore, local $H_2O$-OH miscibility via H-bond is greatly reduced compared to that in the bulk phase, thus ensuring the durability of the non-covalent decoration in solutions.

After clarifying the stability issue, we then prepared the channel of Untreated-M, Ace-M, and EtOH-M in molecular dynamics (MD) simulations and studied monovalent ion transport through them (Fig. 4c and Supplementary Fig. 13). The preparation of solvent-treated channels followed the same process in the lab, and interestingly, stabilized ethanol-treated channels were found having 12.5 times higher decoration coverage than acetone-treated ones. (Supplementary Fig. 14). This agreed well with our DFT- and XPS-based analyzing results, providing a mutual validation of both our simulation and characterization methodologies. As shown in Fig. 4d and Supplementary Fig. 15, monovalent ions could free-pass untreated channels and rapidly accumulate at the channel exit. By contrast, channels stuffed with solvents greatly retarded their permeation depending on the decoration coverage. In particular, the permeation rate of simulated $Na^+$ was reduced by a respective factor of 2.3 in Ace-M and 18 in EtOH-M, in excellent accordance with experimental results. The discrepant ion sieving ability among different channels also displayed a positive correlation with channel decoration degree, which suggests an ion sieving mechanism mainly based on size exclusion.

In summary, 2D membranes possessing selective 1-nm or sub-1-nm channels can potentially extend their separating capability into numerous application settings with appropriate modifications. The easy and universal establishment of non-covalent interactions without radical reactions offers an efficient modifying route that largely maintains membrane integrity and channel alignment. As a proof-of-concept, we demonstrate the possibility of effective and stable non-covalent decoration to $Ti_3C_2T_x$ MXene channel by building up multiple hydrogen bond networks between protic solvents and channel surface groups such as −O, −F, and −OH. More importantly, the angstrom-scale channel width induces the nanoconfinement effect to regulate the distance and orientation of the solvent to channel walls, rendering the generally weak H-bond sufficiently robust to yield improved ion sieving and separating performances. This study proves the feasibility of non-covalent modifications in nanoconfined space, which is promisingly extendable to other appropriately-sized nanochannels of various forms (e.g., porous and mixed matrix) and materials (GO, $MoS_2$, MOF, COF, etc.). It also indicates that stable preferential settling of one species by non-covalent interactions can boost its selectivity against others in a separation process, providing a new perspective to understanding transporting phenomena in 1-nm and sub-1-nm channels. We believe that these methodologies and underlying theories can be applied to develop and modulate membranes used for a broader range of applications including organic solvent nanofiltrations, solvent separations, gas separations, and beyond.

## Methods

### Preparation of MXene materials

MAX parent phase $Ti_3AlC_2$ (400 mesh, >99.0%) was purchased from Kaixi Tech, China. Lithium fluoride (LiF, BioUltra, >99.0%) was purchased from Sigma-Aldrich while hydrochloride acid (HCl, 36%) was obtained from RCI Labs. All chemicals were used without further purification. The etching process was conducted in the same fashion and under similar conditions as described in previous studies but with slightly more concentrated HCl (7 M)[26]. In detail, LiF (1 g) was first mixed into HCl (20 ml) in a 150 ml Teflon container under magnetic stirring at 300 rpm for 5 min. MAX (1 g) powder was then slowly added into the mixture over 5 min to avoid overheating. The temperature of the container was maintained at 35 °C for the next 24 h for sufficient etching. Upon the finishing of the etching, the etched product was

washed via a "centrifugation-dispersion" method using deionized water. Normally the centrifugation was carried out at RCF of 550×*g* (Rotor 12181, Sigma 2-16P) until the pH of the supernatant reaches around 6, and the sediment was re-dispersed mildly sonicated for 10 min to further delaminate few-layered MXene stacks. The sonicated product was finally centrifuged for another 1 h at an RCF of 550×*g* to remove most of the unexfoliated $Ti_3AlC_2$. The concentration of the as-prepared MXene solution was determined based on a UV-vis spectrum "absorbance−concentration" plot (Shimadzu UV-2600 UV−Visible spectrometer, wavenumber from 900 to 200). Generally, after careful collection, MXene dispersion (around 0.7 mg/ml) could be obtained. A chemically inert gas such as Ar was then pumped into the dispersion before storing it at a low temperature (lower than 4 °C).

### MXene membranes preparation

To fabricate laminar membranes, the calculated amount of MXene would be filtrated on a Nylon substrate (47 mm diameter, 0.2 μm pore size) with vacuum assistance. For instance, to prepare a 300-nm thick membrane, a dispersion containing 1.12 mg of MXene would be extracted and diluted with 200 ml of deionized water. As-prepared membranes were dried at room temperature for 24 h before further tests, treatments, and characterizations. The solvent-modified membranes were prepared via a facile "immersion-evaporation" method, where these pristine membranes were immersed in solvents (5 ml) including ethanol (anhydrous, >99.5% Sigma-Aldrich), acetone (ACS reagent, >99.5% Sigma-Aldrich), cyclohexane (anhydrous, 99.5% Sigma-Aldrich), methanol (HPLC, >99.9% Sigma-Aldrich), acetaldehyde (ACS reagent, >99.5% Sigma-Aldrich), n-hexane (HPLC, >97% Sigma-Aldrich) for 2 h. After that, the solvent-treated membrane was placed in a 50 °C oven to let any residual organic molecules evaporate. All untreated and solvent-treated membranes were further dried at room temperature for 12 h before further testing or characterizations.

### Characterizations

The morphology and size distribution of $Ti_3C_2T_x$ nanosheets and membranes was characterized by scanning electron microscope (SEM) on a Nova NanoSEM 450 microscope (FEI, USA), operated with an accelerating voltage of 5 kV, 2.0 spot size. All samples were sputtered with iridium before SEM image taking. More detailed morphology and crystallinity information of single MXene nanosheets were obtained by transmission electron microscopy (TEM) under conventional imaging and diffraction mode, respectively, on an FEI Tecnai G2 T20 microscope operated at an accelerating voltage of 200 kV. The EDX data were collected using an FEI Tecnai G2 F20 FEGTEM operating at 200 kV with a Bruker EDX detector. The height profile was acquired by atomic force microscopy (AFM) using a tapping mode on Bruker Dimension Icon Microscope (Bruker, Billerica, MA, USA). To analyze the crystallographic structure of bulk $Ti_3AlC_2$ and restacked $Ti_3C_2T_x$ membranes, XRD was conducted on a Bruker D2 Phaser Diffractometer with a Cu-Kα radiation source (15 Ma and 40 kV) and the data was analyzed by coupling software DIFFRAC. The examination of MXene surface chemical properties including atomic ratio, atomic chemical states, and functional groups was performed by XPS on a Thermo Scientific Nexsa Surface Analysis System equipped with a hemispherical analyzer and the data were processed either automatically (element analysis) or manually (peak fitting) on a supporting software Avantage. The incident radiation was monochromatic Al Kα X-rays (1486.6 eV) at 72 W (6 mA and 12 kV, 400 × 800 μm). To determine the inner chemical properties of our samples, depth profile elemental analysis was added, which was realized by etching away top MXene layers for hundreds of seconds. Meanwhile, a qualitative and semi-quantitative probing of MXene functional groups was carried out by FTIR on a Perkin Elmer Spectrum Two FTIR Spectrometer with the widest spectral range from 4000 to 400 $cm^{-1}$.

## Ion-sieving performance test

The ion-sieving ability of MXene-based membranes was evaluated by an ion-diffusion experiment. It was carried out using a Teflon-based U-shape apparatus. In a typical permeation test, 15 ml of 0.2 M XCl (X for H$^+$, K$^+$, Na$^+$, Li$^+$. Mg$^{2+}$ and Ca$^{2+}$) solution and DI water were simultaneously injected into different sides as feed and permeate solution, respectively. Magnetic stirring at 350 rpm was used to minimize possible concentration polarization. To figure out the permeation rate of ions through different membranes, the conductivity of the permeates was measured with a LabCHEM Conductivity-pH meter at room temperature. The relation between measure conductivity and permeate salt concentration can be expressed as:

$$C = \frac{\kappa}{\Lambda_m} \quad (1)$$

where $C$ is the to-be-calculated permeate concentration while $\kappa$ and $\Lambda_m$ respectively represent measured conductivity and molar conductivity of the used salt solution. The obtained $C$ can thereby be converted to ion permeation rate ($J_i$, mmol h$^{-1}$ m$^{-2}$) based on:

$$J_i = \frac{VC}{At} \quad (2)$$

## DFT calculations

Calculations are performed by density functional theory (DFT) based on the projected augmented wave (PAW) in the Vienna ab initio simulation package 5.4 (VASP)[33,34]. The Perdew–Burke–Ernzerhof (PBE) form of generalized gradient approximation (GGA) is used as the exchange-correlation function of energy[6]. The DFT-D3 method is used to describe the weak interlayer van der Waals effect[6]. The cutoff energy of the plane wave is set at 550 eV, and the convergence criteria for energy and force are set at 10$^{-5}$ eV and 0.005 eV/Å, respectively. The 3 × 3 × 1 K-point grids in the Brillouin zone are used for structural optimization and electronic structure calculations. Simplified membrane structures are constructed using 2 layers of surface groups (−O, −OH, −F) decorated MXene supercells (30 × 26 × 55 Å$^3$) with an initial interlayer distance of 16.5 Å derived from XRD experiment results. A 25 Å vacuum layer is set to avoid the interaction between the repeated unit cell normal to 2D layers. Over 60 types of conformation of EtOH to MXene have been tested to find out the optimized structure. To simulate the real application in membrane structure, "1-on-1", "1-between-2", and "1-between-3" configurations of decoration molecules are configured. In order to determine the configuration with the lowest energy, the binding energies of the decorated membrane structures are evaluated by the following equation:

$$E_b = E_{total} - E_{Membrane} - E_{decoration} \quad (3)$$

where $E_{total}$, $E_{Membrane}$, and $E_{decoration}$ represent the total energies of the decorated membrane structures, a double-layered MXene membrane, and an isolated decoration molecule (EtOH, cyclohexane, acetone), respectively. A more negative value indicates stronger binding energy. The optimized interlayer spacing of 16.4 Å is larger than the sum of the radii of O, OH, and F groups of MXene, indicating that there is a typical van der Waals interaction between the MXene layers of the membrane sample. Furthermore, the effect of nanoconfinement on its formation by adjusting the ethanol-MXene distance is investigated based on the Climbing Image Nudged Elastic Band (CINEB) method to illustrate the energy profile for EtOH migration path.

## MD simulations

To investigate the ion transport properties, MD simulations with three models were carried out: surface-modified MXMs (MXene channels with a $d$-spacing of 16.5 Å) equilibrated in water, in ethanol, and in acetone, respectively[26]. The MXM repeat unit is originally from the crystal structure of Ti$_3$C$_2$F$_2$, the functional -F groups are randomly replaced by -O- and -OH reaching a ratio of −O-/-F/-OH = 0.47/0.38/0.14[35]. The box size in the solvent equilibrium process is 5.54 nm × 5.33 nm × 3.28 nm. The systems were filled with water/ethanol/acetone molecules through the solvation process. The water molecules were described by the SPC/E model[36]. The universal force field UFF forcefield with $Q$Eq charge is assigned to other molecules through Openbabel and OBGMX codes[19,37–40]. Systems were subjected to multi-step steepest-descent energy minimization followed by CG energy minimization[26]. After that, 100 ns NVT equilibrium simulations were performed. The MXene atoms were frozen in the simulations since the MXene nanosheets were rather rigid. In NVT simulations, the time step is 2.0 fs, and bonds to hydrogen atoms were maintained with the LINCS algorithm[41]. And a constant simulation temperature of 298.15 K was maintained by the V-rescale thermostat[42]. The rcoulomb and rvdw were set as 12 Å. The electrostatic interactions were evaluated using the particle mesh Ewald algorithm. For Na$^+$/Li$^+$/K$^+$ ion transport behaviors, the simulation box is enlarged into 12 nm × 5.33 nm × 3.28 nm with an additional feed region at the left side of the MXene sheet. Salt ions were randomly placed in the feed chamber with a concentration of 0.42 M. 100 ns NVT production simulations were performed after energy minimization. The permeation rate was calculated as the number of cations permeating into the membrane divided by the simulation time. All MD simulations in this work were performed using the GROMACS 2019.6[43]. Simulation results were analyzed and produced using VMD software[44].

## Data availability

The data that supports the findings of this study are available in the article and supplementary information file. The raw data generated during the current study are available from the corresponding authors. Source data are provided with this paper.

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

## Acknowledgements

Y.K. and X.Z. thank Australian Research Council for funding the Industry Transformation Research Hub for Energy-efficient Separation (IH170100009) by which the work is supported. X.Z. thanks to the Australian Research Council for his ARC Future Fellowship (FT210100593). Q.G. thanks research support from ANSTO, and ASCI computing facility. The authors acknowledge the use of instruments, and scientific and technical assistance at the Monash Centre for Electron Microscopy, a Node of Microscopy Australia, and the use of facilities within the Monash X-ray Platform.

## Author contributions

Y.K. and X.Z. raised the conceptualization and designed the research framework, and they also finished the manuscript writing joined by H.W. Y.K. conducted the experiments with the assistance of Y.W. and K.H. and R.X. Y.H., Z.W., W.Z., and Y.C. conducted most characterizations while these parts were also supported by Z.X. T.H. finished Molecular Dynamics simulation and Q.G. covered all the DFT calculation section. All the authors take part in the discussion and reviewing of the project towards its completion.

## Competing interests

The authors declare no competing interests.
