## [Peer Review File · Nature Communications]

Reviewer comments, first round -

Reviewer #1 (Remarks to the Author):

This paper reports on a facile modification of Ti MXene with ethanol for membrane applications. The authors have characterised the membrane well and also proved the application perspective sufficiently. The title emphasise on nano confinement effect, however, it is not described well why it is happening and what is the advantage Etc. Please find my comments on this manuscript:

1. Add reference to the "A typical example is aquatic-used graphene oxide (GO) and MXene channels that show effective width of ~ 0.8 nm" (Line 43-44)
2. "Highly crystallized, monolayered Ti₃C₂T_x nanosheets (Supplementary Fig. 1) with low oxidation degree and small lateral size distribution (Supplementary Fig. 2) were etched from parent MAX phase". Low oxidation degree should be supported by TGA of the material under O₂ atmosphere.
You can go through the reference - Liu, N., Li, Q., Wan, H. et al. High-temperature stability in air of Ti₃C₂T_x MXene-based composite with extracted bentonite. Nat Commun 13, 5551 (2022). <https://doi.org/10.1038/s41467-022-33280-2>
3. As described in Line 83, "-O, -OH and -F groups linked to Ti atoms, and C-F and C-O, presumably around the edge of nanosheets". How did the authors come to the conclusion of edge functionalisation than surface functionalisation. Please explain.
Also provide the elemental mapping of the edge functionalised MXene to further prove the statement.
4. The concentration of MXene (mg/mL) used for the membrane preparation, volume of MXene solution taken, volume of ethanol or cyclohexane impregnated needs to accurately mentioned in the description of Fig S4.
5. SEM images of the membrane with and without solvent impregnation should be done to analyse the morphological changes if any.
6. Background correction of peak fitting should be included in XPS plots
7. The XPS provided is of synthesised MXene or ethanol treated MXene. Please specify. If not the latter, please provide the XPS of the MXene treated with ethanol to understand the change in surface functionalisation of MXene.
8. Please provide the values of transmittance in FTIR and not in arbitrary units. It should be noted that the after-ethanol impregnation the peak corresponding to -OH should have higher intensity which is not observed in Fig 1c
9. An XRD pattern of treated and untreated MXene with scanning angle up to 70 degree should be included to understand structural changes or oxidative changes as a result of prolonged heating as a part of membrane preparation
10. The oxidative stability of ethanol treated Ti₃C₂T_x should be included as well (UV and TGA)
11. In Fig 2b, the hydrated radii of Mg²⁺ is more than Ca²⁺ but permeability of Mg ions are higher than Ca. What could be the reason for this observation?
12. DMSO (polar and aprotic) also is proved to interact with MXene well. May be the studies on how it interacts with MXene should also be included.
13. Has the channel dimension been experimentally evaluated? If not, please include suitable references to prove the channel dimension to be less than 1nm which is requisite for ion separation.
14. How did the authors rule out the possibility of adsorption of positively charged ions through MXenes that have high negative zeta potential values. Please clarify.
15. Separation of solvent molecules requires a channel dimension of less than 2 nm (as stated in Line 32), why did the author choose separation of ions over solvent molecules?
16. Fig. 4f shows that proton transport was also retarded in the ethanol-anchored MXene

channels, albeit by a much smaller extent compared to other salt transport. There is no Fig. 4F in the manuscript.

17. Abstract: Could the authors also mention impedance to 1-nm mass separations due to fouling?

18. Line 48-“The majority of alike modifications” can be changed to “A majority of such modifications”

19. Inconsistencies in sentence construction- Introduction and materials and methods

20. Could the authors please clarify what two-fold implication means in XRD? Perhaps, they mean a change in d spacing? Also, there is a slight change in FWHM, then how do the authors confirm if the hydrated interlayer channel size remained the same regardless of ethanol treatment? The slight variation in FWHM, could indicate a change in d spacing? Also HRTEM images of the materials could be taken to understand if there are any corroborations with XRD.

21. The authors mentioned an improvement in ion sieving efficiency, how can this be correlated to the XRD observation, where the authors state hydrated inter layer channel size remained the same.

22. Also, could the authors estimate the reduction in channel width by studying the width before and after ethanol modification and also taking into account the size of ethanol molecule?

23. Another point that is to be highlighted is the molecular shielding effect. The authors state that the permeation rate is unchanged in 168 hours. Is this performance dependent on flow rate. Also, this flow rate can be imagined from a point of view where ethanol completely covers the material surfaces. Given the already anionic nature of MXene before ethanolic coverage, do the authors anticipate a repulsion between modifying ethanol molecules and surface functionalities. Such a repulsion would lead to non-uniform coverage.

24. In figure 7 the authors have taken 300 nm membrane for studies. Could you please explain why the 100 nm was not taken? Also what is the influence of flow rate on S7 figure a? In addition, in the case of 300 nm thick membranes, it is mentioned that after 0.2 hours of ethanol treatment, further ethanol is not able to permeate the membrane, so in this case, what is the extent of modification of the membrane at 0.2 hours, is it 100% modification or is the modification % lower?

25. The authors have not included a comparison with other similar literature reports (if any). May be a table to this end can answer this question.

26. In the permeation test, 0.2 M HCl was used, which is highly acidic. But the result for this is not included in Fig 2. Could the authors please include this as well.

27. Also, the pH of the solution being passed may affect the affinity for ethanol toward the MXenes membrane. Could the authors please shed some light into this aspect?

Minor comments

1. Please italicise the letter 'x' in Ti₃C₂T_x

2. A lot of emphasis is given to the phenomenon of 'nano-confinement'. Please elaborate the role of nano-confinement in the present context under the section of 'Ion sieving and separating performance of ethanol-treated membranes.

3. JCPDS of MXene synthesised should be provided.

4. Novelty of the work should be included in the introductory and concluding sections.

5. The efficiency of the membrane should well compare and validated with the already reported cation selective membranes based on MXene

6. The membrane is selective to specifically cations and not all ions; appropriate title modification is suggested.

7. In Supplementary fig. 13, why there is a decrease in the number of permeated K⁺ at some points ? What it means?

Reviewer #2 (Remarks to the Author):

This manuscript describes a simple method in regulating ion transport properties of Ti₃C₂T_x MXene membranes. The method is based on solvent treatment using protic solvent which “rearrange” the H-bond network within MXene membranes, providing a long-lasting effect in

tuning the ion transport behavior. The manuscript also attempts to provide an explanation (including experimental data, DFT calculations and MD simulations) on the role of different solvent molecules in the nanochannel of the membrane. However, I believe additional analysis and discussion are needed on this front which will offer further insight to the ion transport mechanism. Nonetheless, I think this manuscript is well-written, and the results and arguments are clearly articulated. I would recommend accepting the manuscript for publication after some minor revisions are made.

My main comment regarding the manuscript is:

1. The main argument of the work is that EtOH enhances the "nanoconfinement effect" within the MXene membrane. It is known that certain polar solvents have a strong affiliation (vs. water) to MXene, which is confirmed by the DFT calculation. While MD simulations reveal some information regarding the ion transport through the membrane. I think the underlying mechanism can be further articulated based on the data presented. To me the effect of EtOH seems to be two-fold: introducing steric effect/size exclusion, and also affecting the ion transport mechanism. The discussion on the latter is missing here.

Figure 2c shows that the activation energy increased considerably after EtOH treatment. It indicated that transport of Na⁺ ion is transitioning from Grotthuss-like (hopping) mechanism (20 kJ/mol) to a (sluggish) transport mechanism (39 kJ/mol). This is an evident that EtOH in the channel is "breaking" the hopping pathway for Na⁺. I think providing some additional data (e.g. Arrhenius plot for proton and other selected cations) may help elucidate this better.

Additional minor comments are as follows:

2. Some section of the introduction has no citations (such as Line 41-51)

3. It would be useful to provide some information regarding the oxidation status of the membrane after test, as MXene is prone to oxidize in aqueous environment. EtOH treatment may be somewhat helpful to the oxidation stability.

4. Solvents such as methanol and n-hexane were also mentioned in the experimental section, but there seems to be no discussion in the main text. The required treatment time (Figure S7) was identified by using Na⁺ ions. Would this change with other ion species? It will be good if results on other ions can be tested to cross compare.

5. The membrane drying time and ethanol treatment time described in the experimental section is confusing (Line 294-298) and does not seem to match with the data reported in Figure S7.

6. What is atomic ratio from XPS? There might be some minor deconvolution issues (likely related to calibration), e.g., C-C (at ~283.2eV) and Ti-C seems to be off?

7. Reference 10 does not seem to provide direct evidence on the claim that "volatile by-products including CO₂ and H₂O will interrupt the channel alignment and cause unwanted enlarged size distribution". I would consider rewording this argument.

8. AFM image is showing the MXene is somewhat oxidized. The rough and raised edges are usually an indication of oxidation. It will be better to replace this AFM image with a cleaner one.

Reviewer #3 (Remarks to the Author):

The authors report on an interesting phenomenon where solvent molecules decorate the MXene surface to enhance sieving properties. In overall, it would be better if more results are provided to understand the sieving mechanism and the practicality of the method is further investigated. I believe this paper can be published after addressing some major revisions below:

Comments

- Authors are encouraged to perform the same experiment and present data throughout the paper using another alcohol (such as methanol) to further prove the influence of hydrogen bonding. In fact, it is mentioned that methanol is used in the methods section, but no data can be seen.

- How does the permeation rate compare in pressurized sieving systems instead of diffusion systems? It is doubtful whether the ethanol adsorption on MXene will remain intact if a stronger flow is applied to the system, as practical applications employ cross-flow filtration.

- In the main text, it is suggested to briefly elaborate the physicochemical reason behind the blue shift of FTIR peaks after insertion of ethanol, instead of just allocating it to references.

- In figure 1b, why does the C-Ti-O and C-Ti-OH peak disappear after ethanol adsorption? As ethanol attaches non-covalently, the two peaks above should still remain on MXene.
- In figure 2d, the 'onset' point for the increase in permeation rate for untreated-M is around 96 hours of testing. It is curious whether this point never occurs for EtOH-M or whether this point arrives in extended testing times. Are the EtOH-M samples still stable for testing after more than 2 weeks?
- Figure 4f in line 140 should be Figure 2f.
- Please comment on the possibility of using this system in organic solvents toward OSN, etc.
- The mechanism of ion sieving seems to be highly influenced by the physical confinement induced by ethanol adsorption. How does the sieving performance of untreated MXene and EtOH-MXene turn out when these films are annealed at higher temperatures above the current 50 degrees? It is well known that the interlayer spacing significantly decreases when temperature above 120~150 degrees C are used to anneal MXene films.

Response to Reviewers' Comments

Reference Number: NCOMMS-22-48007

Title: Non-covalently Decorated Ion-selective MXene Membranes Enabled by Nanoconfinement

Reviewer #1 (Remarks to the Author): This paper reports on a facile modification of Ti MXene with ethanol for membrane applications. The authors have characterized the membrane well and also proved the application perspective sufficiently. The title emphasize on nano confinement effect; however, it is not described well why it is happening and what is the advantage Etc. Please find my comments on this manuscript.

We are grateful for the reviewer's acknowledgement on our previous efforts, and helpful comments to enhance the manuscript quality. Please find below our further work to improve it following your advice.

Q1. Add reference to the "A typical example is aquatic-used graphene oxide (GO) and MXene channels that show effective width of ~0.8 nm" (Line 43-44)

A1: References have been added where necessary. Also, we made a slight change to the description from "~0.8 nm" to "~0.6 to 0.8 nm" to better reflect various real-world situations.

Q2. "Highly crystallized, monolayered Ti₃C₂T_x nanosheets (Supplementary Fig. 1) with low oxidation degree and small lateral size distribution (Supplementary Fig. 2) were etched from parent MAX phase". Low oxidation degree should be supported by TGA of the material under O₂ atmosphere. You can go through the reference - Liu, N., Li, Q., Wan, H. et al. High-temperature stability in air of Ti₃C₂T_x MXene-based composite with extracted bentonite. Nat Commun 13, 5551 (2022). <https://doi.org/10.1038/s41467-022-33280-2>.

A2: We greatly thank the reviewer for recommending a potential examining way on MXene oxidation. After careful reading the provided paper, we unfortunately realized that TGA might not be best way to identify the oxidation status of MXene dispersion mainly because of two reasons: (1) TGA is usually employed to characterize materials' thermal stability and thermal-related features such as anti-oxidation ability mentioned in this paper. However, it does not reveal the exact oxidation degree of the material before the test. This is in line with the concept of the provided reference, where the author conducted TGA to prove the improved anti-oxidation capability of MEB composites from pure MXene, rather than to compare their initial oxidation status; (2) The oxidation degree of MXene is more of a relative concept and so its precise determination via TGA, if ever possible, will rely on a comparison with an oxidation-free MXene nanosheets as reference. Unfortunately, MXene nanosheet synthesis via current methods is always accompanied by oxidation to a certain extent, which make a TGA-based analysis even less likely.

Instead, most studies so far resort to the combination of microscopic evidence to semi-quantitatively examine on the oxidation degree of freshly prepared MXene nanosheets, which involves direct MXene observations under TEM and SEM (*Chem. Mater.* 2017, 29, 4848–4856). As can be seen in updated Supplementary Fig. 1 and our response to Q1 of Reviewer #2, the TEM and AFM images showed even and smooth edge and clear single-crystal TEM diffraction pattern. These features are highly similar to the previously reported MXene nanosheets with low oxidation extent, thus supporting our claim. Please also be noted that although the recommended paper did not directly solve the oxidation issue, we found it very helpful in explaining the EtOH's protective role in improving EtOH-M stability and thus referenced it in the manuscript and Supplemental Fig. 7. Please refer to Q3 of Reviewer #2 for more information.

Q3. As described in Line 83, “-O, -OH and -F groups linked to Ti atoms, and C-F and C-O, presumably around the edge of nanosheets”. How did the authors come to the conclusion of edge functionalization than surface functionalization? Please explain. Also provide the elemental mapping of the edge functionalized MXene to further prove the statement.

A3: We apologize for the misleading sentence on nanosheet chemical composition. We followed the reviewer’s advice to add Energy-dispersive X-ray analysis (EDX) mapping to the $\text{Ti}_3\text{C}_2\text{T}_x$ nanosheet

under TEM, as shown below and in updated Supplementary Fig. 5. It can be seen that all the main elements including Ti, C, O and F (H is undetectable with current TEM EDX mapping precision) are evenly distributed and mixed, and most functional groups are located on the basal plane of the nanosheet. The original sentence in the manuscript is revised accordingly.

Supplementary Fig. 5:

Supplementary Fig. 5. X-ray photoelectron spectroscopy (XPS) spectra of MXene membranes including (a) O1s and (b) F1s. TEM image of MXene (c) and associated energy Dispersive X-ray (EDX) microanalysis.

...Energy Dispersive X-ray (EDX) microanalysis in selected area (c) show that the main composition elements of MXene including Ti, C, O and F are evenly distributed across its whole surface. This indicates that functional groups attached to the MXene are also well spread not only in the basal plane, but also along the edge of MXene nanosheets.

Manuscript:

(Line 82-85) In accordance with previous studies, X-ray photoelectron spectroscopy (XPS) and energy-dispersive X-ray (EDX) analysis revealed the membrane chemical composition, comprising -O, -OH and -F groups linked to Ti and C atoms across the basal plane and around the edge of nanosheets (Fig. 1 a, b, and Supplementary Fig. 5)^{15,16}

Q4. The concentration of MXene (mg/mL) used for the membrane preparation, volume of MXene solution taken, volume of ethanol or cyclohexane impregnated needs to accurately mentioned in the

description of Fig S4.

A4: We have elaborated the experimental section to include the above-mentioned details, and the updated version is as below and in the “Preparation of Pristine and Solvent Modified MXene Membranes” section. Please be noted that the “inconsistent sentences” as mentioned in Q19 remain here as these data have been added into the manuscript as requested by other reviewers.

(Line 301-314) Generally, after careful collection, MXene dispersion of around 0.7 mg/ml can be obtained... To fabricate laminar membranes, calculated amount of MXene would be filtrated on Nylon substrate (47mm diameter, 0.2 μm pore size) with vacuum assistance. For instance, to prepare a 300-nm thick membrane, dispersion containing 1.12 mg of MXene would be extracted and diluted by 200 ml deionized water. As-prepared membranes were dried in room temperature for 24 hr before further tests, treatments and characterizations. The solvent-modified membranes were prepared via a facile “immersion-evaporation” method, where these pristine membranes were immersed in 5 ml of solvents including ethanol (anhydrous, >99.5% Sigma-Aldrich), acetone (ACS reagent, >99.5% Sigma-Aldrich), cyclohexane (anhydrous, 99.5% Sigma-Aldrich), methanol (HPLC, >99.9% Sigma-Aldrich), acetaldehyde (ACS reagent, >99.5% Sigma-Aldrich), n-hexane (HPLC, >97% Sigma-Aldrich) for 2 hours.

Q5. SEM images of the membrane with and without solvent impregnation should be done to analyze the morphological changes if any.

A5: As suggested, we took and compared the digital photos, surface and cross-sectional SEM images of both Untreated-M and EtOH-M as shown below and in updated Supplementary Fig. 3. These photos reveal that (1) EtOH treatment does not cause any discernable visual difference across the membrane and its substrate; (2) Membrane surface morphology largely remain unchanged after EtOH treatment under SEM either in low resolution (typical ridge-and-valley structure) or high resolution (flatter surface); (3) EtOH treatment hardly alters the layered structure of the membrane. Therefore, it is reasonable to conclude that solvent impregnation can rarely results in obvious morphological changes.

Q6. Background correction of peak fitting should be included in XPS plots.

A6: Please see updated Figure 1 and Supplemental Fig. 5 for XPS plots with background correction.

Q7. The XPS provided is of synthesized MXene or ethanol treated MXene. Please specify. If not the latter, please provide the XPS of the MXene treated with ethanol to understand the change in surface functionalization of MXene.

A7: The XPS spectra provided in original manuscript is for Untreated-M. We followed the reviewer's suggestion to redo XPS for the comparison between Untreated-M and EtOH-M, which is shown as below. It can be seen that the introduction of ethanol leads to no emergence of new peaks and negligible shifts of current peaks (<0.1 eV) in Ti 2p, C 1s, O 1s and F1s spectra, thus denying the likelihood of obvious surface chemical modification caused by ethanol treatment.

The only changes that occur after ethanol treatment are the minor increase of C-O (C 1s, 286.1 eV) from 0.67% to 1.24%, and slight relative ratio rearrangement of O 1s species due to the introduction

of ethanol hydroxyl groups. This is reasonable considering the mild treatment conditions in our experimental protocol, where the drying temperature (50 °C) is way too low to initiate any chemical reactions or defunctionalisation. Please also be noted that the above data are only for comparison purposes here following the reviewer’s request, but not included in the updated manuscript. This is because even if we stick rigidly to the same MXene synthesis protocol, it is hard to produce materials with identical physicochemical properties every single time. Since all the theoretical analysis and calculations are based on previous XPS data, we decided to keep them in the manuscript to avoid data inconsistency.

Q8. Please provide the values of transmittance in FTIR and not in arbitrary units. It should be noted that the after-ethanol impregnation the peak corresponding to -OH should have higher intensity which is not observed in Fig 1c.

A8: We re-baselined the original spectra and created the two spectra with same Y-axis range to obtain an updated FTIR figure as below (left). As can be seen, EtOH-M does generate a -OH peak with increased intensity, broader peak area and shifted peak position, all indicating the formation of H-bond at this region. However, we will not be able to use the increased peak intensity of a proof of H-bond for two main reasons: (1) FTIR normally functions as a semi-quantitative characterization tool. Compared to minor changes to peak intensity, the emergence, vanish or shift of peaks conveys more accurate and convincing information from the sample; (2) In addition to EtOH adsorption, the -OH peak intensity can be also influenced by the local membrane thickness and functionality distribution of the tested area (a very small spot). Therefore, in the updated figures, we focused on the shift of MXene characteristic peaks and the appearance of -CH₃ and -CH₂-, which are intrinsic to EtOH.

Also, we provided relative transmittance instead of absolute transmittance value for updated Y-axis because the latter will cause a severe legibility issue to the figure. Please be noted this presentation method has been widely used in previous studies (*Nano Lett.* 2021, **21**, 3495–3502, *Nature Nanotechnology* **16**, 331–336 (2021)). Corresponding revisions have been made to updated Figure 1c

and in the manuscript in combination with Q3 of Reviewer #3, which is also shown as below.

(Line 85-93) These characteristic groups were also detected in Fourier-transform infrared spectroscopy (FTIR), but with clear red shift at some peaks after ethanol treatment (Fig. 1c). Two major peaks assigned to -OH (346 cm^{-1}), and Ti-O (626 cm^{-1}) moved to lower wavenumber of 3410 and 606 cm^{-1} ¹⁷⁻¹⁹. This is largely because of the hydrogen bond formed between them and the -OH from the inserted ethanol, which causes the electron cloud density change and elongation of involving bonds. Consequently, the strength of the bonds will be decreased to show lower stretching vibration frequency in FTIR²⁰⁻²². Also, two joint peaks emerging at 2920 and 2850 cm^{-1} , which are normally attributed to -CH₃/-CH₂-, also proves the successful insertion of EtOH molecules into MXene membranes...

Q9. An XRD pattern of treated and untreated MXene with scanning angle up to 70 degrees should be included to understand structural changes or oxidative changes as a result of prolonged heating as a part of membrane preparation.

A9: XRD scanning of both Untreated-M and EtOH-M in wider angle range ($4\text{-}70^\circ$) was conducted as per the reviewer's request, and the spectra were compared as below and in updated Supplementary Fig. 6. It can be seen that Untreated-M and EtOH-M show almost identical pattern in both dry and wet conditions, possessing an outstanding (002) peak at 6.5° and at 5.4° , respectively. This proves that 2 hr heating at 50°C did not cause any obvious structural or oxidative changes in EtOH-M. In fact, it is believed that 80°C (at vacuum) is the thermal threshold for any substantial change to happen (*ACS Nano* 2019, 13, 10535–10544).

hydration. Meanwhile, it is worth noting that, in both dry and hydrated conditions, the rest of XRD spectra (7° to 70°) for EtOH-M is also almost identical to that for Untreated-M. This implies that moderate heating of EtOH-M at 50°C will not cause any substantial structural or chemical changes to the membranes.

Q10. The oxidative stability of ethanol treated $\text{Ti}_3\text{C}_2\text{T}_x$ should be included as well (UV and TGA)

A10: The relative oxidative stability of EtOH-M to Untreated-M was proved by a commonly accepted method based on membrane conductivity test, and was explained in updated Supplemental Fig. 7. Please refer to Q3 of Reviewer #2 for detailed experimental and mechanism explanation.

Q11. In Fig 2b, the hydrated radii of Mg^{2+} is more than Ca^{2+} but permeability of Mg ions are higher than Ca. What could be the reason for this observation?

A11: Considering their error and the very close hydrated size of Mg^{2+} (8.6 Å) and Ca^{2+} (8.4 Å), we would say that the permeation rates of Mg^{2+} and Ca^{2+} are in the same range. Also, the permeation of ions in confined channels is dependent on a number of factors, including size, surface functionality. Hence, it is not a surprise that the permeation rate of Mg^{2+} can sometimes be slightly higher than that of Ca^{2+} . In fact, such phenomenon was also in very recent study of MXene-based membranes (*Nature Sustainability* **3**, 296–302 (2020)).

Q12. DMSO (polar and aprotic) also is proved to interact with MXene well. May be the studies on how it interacts with MXene should also be included.

A12: We appreciate the reviewer's suggestion in treating MXene membrane with another unique solvent and so we prepared DMSO-M following exactly the same experimental protocol for other solvents. Very sadly, our attempts to test its ion-sieving/selective performance were not successful due to the exceptional dissolving ability of pure DMSO on the substrate we used (we tried Nylon and PAN, Polyacrylonitrile). Over a short period of immersion, the top surface of nylon was dissolved, leading to the peeling off or disintegration of MXene membrane from the substrate and making further performance testing impossible.

While it is a shame that we could not directly investigate MXene-DMSO interaction, we instead chose acetaldehyde (MeCHO) to treat MXene membrane, and re-evaluated how these polar yet aprotic solvents interact with and decorate MXene channels. Combined with other reviewers' similar questions, detailed experimental data and discussion can be found in our response to Q1 of Reviewer #3.

Q13. Has the channel dimension been experimentally evaluated? If not, please include suitable references to prove the channel dimension to be less than 1nm which is requisite for ion separation.

A13: The most effective and universally acknowledged characterization tool to evaluate the channel dimension (or size) of 2D lamellar membranes is XRD, and our study followed the same test and

analysis protocol. Please refer to your Q20 to 22 where we provide detailed explanations to all your concerns with channel size.

Q14. How did the authors rule out the possibility of adsorption of positively charged ions through MXene that have high negative zeta potential value? Please clarify.

A14: It is true that adsorption of ions can happen in membrane-based operations due to electrostatic attraction. However, this process is commonly considered occurring during the very early stage of the test and reaching its equilibrium quickly.

To prove this viewpoint, we actively monitored Na^+ permeation through both 300-nm thick Untreated-M and EtOH-M in the first 5 mins using conductivity meter. Before exhibiting the typical “testing time-conductivity” linear relation, both Untreated-M and EtOH-M had a “dormant period” where the conductivity reading remained unchanged, a phenomenon largely attributed to ion adsorption. More importantly, such dormant period existed for less than 1 min. Considering that tests in our study could run up to 12 hours, it is reasonable to conclude that the impact of ion adsorption to the overall permeation rate is negligible.

Q15. Separation of solvent molecules requires a channel dimension of less than 2 nm (as stated in Line 32), why did the author choose separation of ions over solvent molecules?

A15: As the reviewer expected, the effective channel size of our Untreated-M is less than 0.8 nm (Please see Q20 for details) and this value falls into the range of solvent separation. However, we chose to focus on ion separation mainly because (1) this channel size is close to that of smallest salt ions (e.g., K^+ 0.66 nm and Na^+ 0.72 nm) and so a minor decoration to the channel can yield potentially high sieving and separating performance; (2) compared to hydrated ions who possess isotropic size, the size of solvent molecules can vary depending on specific molecular direction and this may make organic solvent less responsive to size-based separation mechanism; (3) as proved in this work, the transport of certain organic solvents (e.g., ethanol) can be largely affected by not only the channel size but also

its surface chemistry, potentially leading to unusual transporting phenomena.

Therefore, we started our research from aqueous system. However, we are also working on solvent-related separation on similar systems at the moment.

Q16. Fig. 4f shows that proton transport was also retarded in the ethanol anchored MXene channels, albeit by a much smaller extent compared to other salt transport. There is no Fig. 4F in the manuscript.

A16: We apologize for the typo here. This was meant to be Fig. 2f and it has been corrected to what it should be.

Q17. Abstract: Could the authors also mention impedance to 1-nm mass separations due to fouling?

A17: We definitely agree with the reviewer that fouling could be a factor that discourages 1-nm mass separations from being achieved. However, considering the main research focus of this work and more importantly, the abstract word limit (150 words) for the journal, we unfortunately are unable to introduce more fouling-related contents into the abstract, and we apologize for this.

Q18. Line 48- “The majority of alike modifications” can be changed to “A majority of such modifications”

A18: We have changed the vocabulary at the right spot as requested.

Q19. Inconsistencies in sentence construction- Introduction and materials and methods.

A19: We apologize for the inconsistencies showing up in these sections, and we believed that the reviewer was mainly referring to the “methanol-treated” and “hexane-treated” experiments that were missing from the previous manuscript. Please be noted these have been corrected in the revised manuscript with extra data added. Please see Q1 of Reviewer #3 for relevant information.

Q20. Could the authors please clarify what two-fold implication means in XRD? Perhaps, they mean a change in d spacing? Also, there is a slight change in FWHM, then how do the authors confirm if the hydrated interlayer channel size remained the same regardless of ethanol treatment? The slight variation in FWHM, could indicate a change in d spacing? Also, HRTEM images of the materials could be taken to understand if there are any corroborations with XRD.

A20: The two-fold implications obtained from XRD spectra were interlayer channel size and size distribution, respectively identified by peak position 2θ and FWHM. This is because 2D stacking membranes consist of crystalline 2D material layers arranged in a repeating pattern, so the distance between these repeating layers, namely d -spacing, is obtainable by applying Bragg's law $n\lambda = 2d \sin\theta$ at membrane stacking direction. Meanwhile, FWHM is a measure of crystallinity degree and thus is interpreted as the order of membrane stacking structure or size distribution. (e.g., *Nature* **550**, 380–

383 (2017) & *Nature Materials* **16**, 1198–1202 (2017)).

Our study followed the same protocol and calculated the d -spacing of hydrated Untreated-M channels as 16.5 Å from XRD (002) peak. Considering that d -spacing measures the “center-to-center” distance between two neighboring layers, the “effective” channel size should be calculated by subtracting $\text{Ti}_3\text{C}_2\text{T}_x$ thickness (around 8.8 Å) from the d -spacing (Figure above). This answers Q13 on how we identify Untreated-M channel size as 7.7 Å. On this basis, the almost unchanged (002) peak position and FWHM for EtOH-M proves two points. (1) The EtOH decoration does not cause a d -spacing change, showing that one EtOH molecule is only attached to one side of the channel rather than bridging both sides; (2) FWHM does not indicate d -spacing change, but d -spacing distribution. The slight change thus means that EtOH decoration posed very minor disruptions to the original stacking structure, which is an advantage of non-covalent decoration compared to disruptive covalent ones.

While HRTEM may present similar channel size information, it unfortunately does not work in our circumstance as the instrument cannot deal with water-containing membrane samples. Even if it can, the hydrated membranes will be dried in high TEM vacuum, thus presenting misleading information.

Q21. The authors mentioned an improvement in ion sieving efficiency, how can this be correlated to the XRD observation, where the authors state hydrated inter layer channel size remained the same.

A21: To help the reviewer understand in an easier way, we drew a schematic graph in Q20. Indeed, the XRD for Untreated-M and EtOH-M possess almost identical d -spacing. However, like we explained in Q20, this does not mean their effective channel size is the same. With ethanol decorated onto the channel wall, the steric hindrance will reduce the effective size of EtOH-M channels and thus lead to enhanced ion-sieving and ion-selective efficiency.

Q22. Also, could the authors estimate the reduction in channel width by studying the width before and after ethanol modification and also taking into account the size of ethanol molecule?

A22: As stated in Q21, the current model has already considered the reduced effective channel width due to ethanol molecules. However, to precisely quantify such reduction is very tough and tricky as the ethanol molecules at different spots can have varied folding and rotation degree, thus occupying the channel by different extent. For this reason, we only qualitatively explain the decoration effect in our manuscript, as highlighted below.

(Line 112-114) Once decorated with ethanol, it is expected that part of the channel is occupied to narrow down the effective channel width and obstruct the passage of ions.

Q23. Another point that is to be highlighted is the molecular shielding effect. The authors state that the permeation rate is unchanged in 168 hours. Is this performance dependent on flow rate? Also, this flow rate can be imagined from a point of view where ethanol completely covers the material surfaces. Given the already anionic nature of MXene before ethanolic coverage, do the authors anticipate a repulsion between modifying ethanol molecules and surface functionalities. Such a repulsion would lead to non-uniform coverage. (explain DFT in some details)

A23: In this study, membranes were tested in diffusion mode as shown in Fig 2a. The ion permeation rate of the Ethanol-treated membrane is unchanged in 168 hours under our current operation conditions. In diffusion mode, water and ion transport in opposite directions inside nanochannel. In principle, Ion permeation rate is impacted by water flow rate. Higher water flow rate often leads to lower ion permeation rate. However, this normally occurs only when the salt solution concentration is sufficiently enough to create enough osmotic pressure across the membranes (e.g., for NaCl the typical concentration is 1.0 mol/L). Considering that our solution concentration is 0.2 mol/L, the flow rate of water can be reasonably omitted.

DFT analysis (Figure 3d) shows that, more often than not, there are attractions between ethanol molecules and surface functionalities of MXene nanosheets, particularly between ethanol and -OH groups. More importantly, considering that MXene surface is normally highly functionalized due to its etching-based synthesis method, EtOH molecules are supposed to interact with more than one functionality during the modification process. On this basis, our further DFT results (Figure 4a) show that the interactions between modifying EtOH and functionalities are always attractive, though with different strength. Despite these attractions, we do agree with the reviewer that ethanol coverage is non-uniform. This is because the coverage is dependent on the functionality distribution on MXene surface, which is usually considered random and non-uniform.

Q24. In figure 7 the authors have taken 300 nm membrane for studies. Could you please explain why the 100 nm was not taken? Also, what is the influence of flow rate on S7 figure a? In addition, in the case of 300 nm thick membranes, it is mentioned that after 0.2 hours of ethanol treatment, further ethanol is not able to permeate the membrane, so in this case, what is the extent of modification of the membrane at 0.2 hours, is it 100% modification or is the modification % lower?

A24: The reason why 300 nm membranes were used in the testing is because its high mechanical strength. Moreover, we realized that 100-nm thick membranes could have uneven thickness and even defects across their surface. As stated in our response to Q23, higher water flow rate often leads to lower ion permeation rate in diffusion mode testing.

For the time-being, it is indeed very challenging to precisely quantify the extent of modification of the membrane at 2 hours, but the estimate on modification extent for saturated channels is possible.

According to our DFT results (Figure 4a), the modification extent should be theoretically 100% if we consider both physically and chemically adsorbed EtOH. However, if we only consider chemically stable EtOH modification, then the modification extent is 27% (Supplemental Figure 11). In an ideal situation, that is if EtOH can freely permeate throughout the membrane, these value should be the membrane modification extent.

Q25. The authors have not included a comparison with other similar literature reports (if any). May be a table to this end can answer this question.

A25: Please find updated Table S1 for a modification efficacy comparison of the proposed method to recently reported 2D channel modification methods, including covalent and physical ones. Such comparison is based on ion (Na^+ , K^+ and Li^+) sieving performance, and the performance enhancement after modification. Although this is not an exhaustive table, it still proves that our proof-of-concept non-covalent decoration can be as effective as other ways. Accordingly, the manuscript has been revised as below:

(Line 115-118) Such improved ion sieving performance is comparable to that of previously reported covalently- and physically-modified 2D membrane channels, suggesting similar decoration efficacy via non-covalent methods. (Supplemental Table 1).

Q26. In the permeation test, 0.2 M HCl was used, which is highly acidic. But the result for this is not included in Fig 2. Could the authors please include this as well.

A26: The H^+ permeation rate was actually included as the inset to Fig. 2f along with H^+/M^{n+} selectivity. However, we realized the font of the inset caption might be too small to attract attention. We apologize for it and have enlarged it to improve legibility. While to include this dataset into Fig. 2b is also a good option, we prefer to keep it at its original position because Fig. 2b is mainly devoted to exhibiting the superior sieving/rejecting ability of our membranes towards metal cations. Although H^+ transport is also moderately retarded after EtOH treatment, this limited sieving efficiency does not match the aim of Fig. 2b. Meanwhile, Fig. 2f is served as an outlook for the membrane's potential for future ion selective application, having H^+ permeation rate data presented here may be easier for our readers to grab the idea.

Q27. Also, the pH of the solution being passed may affect the affinity for ethanol toward the MXene membrane. Could the authors please shed some light into this aspect?

A27: Theoretically, the H-bond strength solely depends on the electronegativity of involved atoms or groups and the modification of pH and ionic strength (IS) itself will have very limited impact. However, in the current context, the extreme acidic environment is supposedly to enhance the H-bond strength in two possible ways: (1) excessive H^+ in the channel will largely discourage the deprotonation of -OH ($-\text{OH} = -\text{O}^- + \text{H}^+$) on MXene, allowing more -O-H to engage in H-bond establishment; (2) highly

acidic environment will decrease effective surface potential (Electro-attraction) of MXene and thus lead to narrower d -spacing. This will further reinforce nanoconfinement to boost H-bond strength. (I have actually finished a set of experiments to prove membrane's stability in acidic media, and this will put on here once the figure is ready).

Minor comments

Q1. Please italicize the letter 'x' in $Ti_3C_2T_x$

A1: Please be noted all the subscript letters "x" have been italicized as suggested.

Q2. A lot of emphasis is given to the phenomenon of 'nano-confinement'. Please elaborate the role of nano-confinement in the present context under the section of 'Ion sieving and separating performance of ethanol-treated membranes.

A2: In fact, the roles of nano-confinement is revealed and discussed in details in section 4 (Mechanisms underlying channel stability and ion-sieving ability) based on DFT result analysis. In general, its roles can be understood as: (1) limiting MXene-EtOH distance. The sub-1-nm MXene channel can limit MXene-EtOH distance to below 3.5 Å for much stronger H-bond, which ensures much more robust EtOH decoration than in bulk solution for improved and stable performance; (2) limiting EtOH's molecular orientation. This enables constant EtOH facing towards MXene surface with its OH, reducing its interaction with H₂O to further improve decoration stability.

Since section 4 is constructed to explain the experimental phenomena in section 2 (membrane performance), we accordingly changed the title of section 4 to "**Nanoconfinement-enabled channel stability and ion-sieving ability.**" to better serve its purpose.

Q3. JCPDS of MXene synthesized should be provided.

A3: While we tried our best to sort out the JCPDS of MXene in ICDD (International Centre for Diffraction Data), no such data could be found mostly because that MXene ($Ti_3C_2T_x$) can be synthesized using different methods, agents and formulas, each leading to potential unique XRD patterns and thus a standardized pattern is very hard to be defined. However, we can tell the XRD of our $Ti_3C_2T_x$, largely assembles to that of previous studies that adopted similar synthesis method and etching agents (*Nature Sustainability* **3**, 296–302 (2020)).

Q4. Novelty of the work should be included in the introductory and concluding sections.

A4: Please see the updated introduction (last paragraph) and conclusion where we have made according revision.

Q5. The efficiency of the membrane should well compare and validated with the already reported cation selective membranes based on MXene.

A5: Please see Major Q25 for relevant information, as we realized these two questions are highly similar. However, we mainly focus on cation-sieving membranes because (1) the testing parameters cation-sieving membranes are easier to compare; and (2) paper number of H⁺/Mⁿ⁺ MXene membranes is currently low.

Q6. The membrane is selective to specifically cations and not all ions; appropriate title modification is suggested.

A6: We agree with the reviewer's suggestion and slightly modify the title to "Non-covalently Decorated Ion-sieving MXene Membranes Enabled by Nanoconfinement". This is because the manuscript is more about proposing and rationalizing a the nanoconfinement-induced membrane decoration strategy, which leads to very decent ion sieving performance. While H⁺/Mⁿ⁺ selectivity is also tested, more efforts are needed for further performance enhancement and mechanism exploration.

Q7. In Supplementary fig. 13, why there is a decrease in the number of permeated K⁺ at some points? What it means?

A7: The intermittent number decrease of K⁺ is the synergic result of its Brownian motion in simulation process and our statistic analyzing method. To mimic real-world situation, all ions including K⁺ were set free in the MD simulation box to allow a constant Brownian state when permeating MXene channels. Therefore, while these K⁺ streamed into the channel under the chemical potential difference (generated by concentration gradient), a small proportion of them could indeed flow back at certain moments. Meanwhile, statistical analysis on ion permeation in MD followed an established method in previous works (Ref.) by counting the number of ions passing through the channel boundary per nanosecond. In our MD, such counting was conducted at a high frequency (40 times over 100 ns) to ensure data reliability. As a result, it is reasonable that ion backflow could be "occasionally" captured at some counting points to exhibit a dip observed in Fig. 13. Despite these occasional dips, the collected data still clearly show an overall increase over 100 ns and agree well with our experimental results.

Reviewer #2 (Remarks to the Author): This manuscript describes a simple method in regulating ion transport properties of Ti₃C₂T_x MXene membranes. The method is based on solvent treatment using protic solvent which "rearrange" the H-bond network within MXene membranes, providing a long-lasting effect in tuning the ion transport behavior. The manuscript also attempts to provide an explanation (including experimental data, DFT calculations and MD simulations) on the role of different solvent molecules in the nanochannel of the membrane. However, I believe additional analysis and discussion are needed on this front which will offer further insight to the ion transport mechanism. Nonetheless, I think this manuscript is well-written, and the results and arguments are

clearly articulated. I would recommend accepting the manuscript for publication after some minor revisions are made. My main comment regarding the manuscript is:

Reply: We greatly appreciate the positive and instructive comments from the reviewer, and devoted additional endeavors into addressing the reviewer’s questions in transport mechanism explanation and membrane characterizations.

Q1. The main argument of the work is that EtOH enhances the “nanoconfinement effect” within the MXene membrane. It is known that certain polar solvents have a strong affiliation (vs. water) to MXene, which is confirmed by the DFT calculation. While MD simulations reveal some information regarding the ion transport through the membrane. I think the underlying mechanism can be further articulated based on the data presented. To me the effect of EtOH seems to be two-fold: introducing steric effect/size exclusion, and also affecting the ion transport mechanism. The discussion on the latter is missing here.

Figure 2c shows that the activation energy increased considerably after EtOH treatment. It indicated that transport of Na⁺ ion is transitioning from Grotthuss-like (hopping) mechanism (20 kJ/mol) to a (sluggish) transport mechanism (39 kJ/mol). This is an evidence that EtOH in the channel is “breaking” the hopping pathway for Na⁺. I think providing some additional data (e.g. Arrhenius plot for proton and other selected cations) may help elucidate this better.

A1: We thank the reviewer for the insightful advice and we followed it to calculate the activation energy change of H⁺ transport from its Arrhenius plot before and after EtOH treatment, as shown below:

Indeed, as can be seen in H⁺ plot, H⁺ transport in Untreated-M channels were mostly undisturbed, requiring a fairly low activation energy E_a of 11.3 kJ/mol (0.12 eV). This well fell between 0.1 to 0.4 eV, the energy range commonly acknowledged for Grotthuss-like, or hopping, transport mode (*Anal. Chem.* 2014, **86**, 9362–9366). This is reasonable considering the appropriate effective Untreated-M channel size (7.7 Å to include two layers H₂O molecules) and its hydrophilic surface, which provides a continuous H-bond network across the majority of the channel for H⁺ to hop. In EtOH-M, however, higher E_a of 23.1 kJ/mol was observed, indicating H⁺ partial transition from hopping to sluggish

transporting mode. Combining with our DFT results (Fig 4a), we reckon this is because that the decorated EtOH molecules normally have their -OH facing toward the nanosheet and -C₂H₅ facing toward the channel. The replacement of original hydrophilic -OH and -O by more hydrophobic -C₂H₅ at these spots will break down the H-bond network and thus increase H⁺ transport resistance.

While the similar trend occurs to Na⁺ as well, we are not fully confident if the above transporting mode transition also applies. This is because whether metal ions like Na⁺ can “hop” is still in deep debate. However, it is obvious that the appearance of more hydrophobic in the channel -C₂H₅, besides posing steric effect, also causes extra barrier for Na⁺ dehydration process to further slow down its transport. The above data and mechanism analysis have been added in the reconstructed Figure 2 and corresponding paragraphs.

(Line 148-153) The enhanced selectivity was rationalized by Arrhenius plot of temperature against ion permeation rate (Fig. 2f). Although both ions experienced higher energy barrier in EtOH-M than in Untreated-M, the activation energy increase for Na⁺ (from 20.8 to 39.1 kJ mol⁻¹) is obviously more pronounced than that for H⁺ (11.3 to 23.1 kJ mol⁻¹). This implies that the decorated EtOH molecules created substantially larger barrier for Na⁺ to dehydrate while they only moderately interfered with the “hopping” (Grotthuss-like transport) of H⁺.

Additional minor comments are as follows:

Q2. Some section of the introduction has no citations (such as Line 41-51)

A2: A series of references have been added to relevant contents in these lines.

Q3. It would be useful to provide some information regarding the oxidation status of the membrane after test, as MXene is prone to oxidize in aqueous environment. EtOH treatment may be somewhat helpful to the oxidation stability.

A3: We were intrigued by the potential effect of EtOH in moderating MXene oxidation as much as our reviewers, and thus conducted a proof-of-concept experiment based on membrane conductivity (σ) testing, which is consider a reliable indicator of solid MXene oxidation degree (lower σ implies higher oxidation, *Matter* **1**, 513–526, August 7, 2019 & *npj 2D Mater Appl* **3**, 8 (2019)). The testing was carried out using a homemade assembly as shown below, where the Untreated-M or EtOH-M was firmly clamped and attached to conductive Cu foils for easy and stable measurement. The assembly was then kept in water for up to 10 days but taken out and thoroughly dried for resistance (R) measurement at the 1st, 3rd, 7th, and 10th day, and σ can be calculated by the equation below:

$$\sigma = \frac{L}{R \cdot A} \quad (L \text{ and } A \text{ respectively represents tested membrane length and cross-sectional area)}$$

Since L and A remained the same throughout the experiment, σ at different days could be simplified as $\sigma = 1/R$ and thus normalized for comparison purpose. It can be seen that both membranes experienced conductivity loss over the 10-day period, indicating ongoing MXene oxidation. However, such loss in Untreated-M was significantly faster, with conductivity plummeting to below 10% in 3 days, than in

EtOH-M whose conductivity could maintain above 50% of its initial value. Although a fully quantitative link can yet be established between conductivity loss and ion permeation increase, we believe that the above comparison well verifies the enhanced resistance of EtOH-M against oxidation.

We attributed the attenuated oxidation to the “shielding effect” of the absorbed EtOH molecules, which posed a steric hindrance to stop O_2 and H_2O from attacking MXene surface. Accordingly, the above methodology and reasoning are added as below and in updated Supplemental Fig. S7. It should be noted, however, that we did not directly use the membranes in ion-sieving performance testing for conductivity comparison. This was because that peeling these membranes from the device could cause minor damages to the membrane periphery and made the conductivity measurement inaccurate.

Supplementary Fig. 7. (a) Comparison of the anti-oxidation ability of Untreated-M and EtOH-M in aqueous conditions by (b) conductivity test. The normalized results are shown in (c).

Unwanted oxidation of MXene materials including $Ti_3C_2T_x$ is a universal problem in either dispersion or solid form (e.g., layered membranes). This happens when reactive H_2O and O_2 attack the Ti-C bonds of $Ti_3C_2T_x$ nanosheets, resulting in their de-functionalization and breaking. However, ethanol molecules attached onto the $Ti_3C_2T_x$ will prevent the direct contact of the material with reactants, which creates a “shield” to largely slow down the oxidation process (Fig. S6a). To prove this proposed mechanism, we designed a proof-of-concept experiment based on membrane conductivity (σ) testing, which is consider a reliable indicator of solid MXene oxidation degree (lower σ implies higher oxidation). The testing was carried out using a homemade assembly where

the Untreated-M or EtOH-M was firmly clamped and attached to conductive Cu foils for consistent measurement (Fig. S6b). Both assemblies were kept in water, taken out and thoroughly dried for resistance (R) measurement at the 1st, 3rd, 7th, and 10th day, and σ can be calculated by the equation below:

$$\sigma = \frac{L}{R \cdot A}$$

where L and A respectively represents tested membrane length and cross-sectional area, and further simplified as:

$$\sigma = \frac{1}{R}$$

, since L and A remained unchanged throughout the experiment. On this basis, σ measured at each testing point was normalized against its initial σ_0 and plotted in Fig. S6c. It can be seen that both membranes experienced conductivity loss over the 10-day period, indicating ongoing MXene oxidation. However, such loss in Untreated-M was significantly faster, with conductivity plummeting to below 10% in 3 days, than in EtOH-M whose conductivity could maintain above 50% of its initial value. Although a fully quantitative link can yet be established between conductivity loss and ion permeation increase, we believe that the above comparison well verifies the enhanced resistance of EtOH-M against oxidation.

Q4. Solvents such as methanol and n-hexane were also mentioned in the experimental section, but there seems to be no discussion in the main text. The required treatment time (Figure S7) was identified by using Na^+ ions. Would this change with other ion species? It will be good if results on other ions can tested to cross compare.

A4: We apologize for the inconsistency between the experimental section and main text, as the data for methanol and n-hexane treated membranes were collected to reassure the feasibility of the proposed method but not included in the previous manuscript. Encouraged by all the three reviewers, we brought these data back in the updated manuscript to help validate our methodology and theory in this work. Please refer to Q1 of Reviewer #3 for more detailed discussion and data.

In terms of the treatment time-dependent ion permeation tests, we followed the reviewer's suggestion and repeated the same experiment on K^+ and Li^+ whose data are shown as below and in updated Supplemental Figure 8. Meanwhile, the additional tests were applied to thickness-dependent ion permeation to ensure data consistency. It can be clearly seen that the permeation rate "plateau" shows for Na^+ are observed in K^+ and Li^+ cases as well.

Q5. The membrane drying time and ethanol treatment time described in the experimental section is confusing (Line 294-298) and does not seem to match with the date reported in Figure S7.

A5: We apologize for the confusing information here. We have revised the section as below.

(Line 309-314) The solvent-modified membranes were prepared via a facile “immersion-evaporation” method, where these pristine membranes were immersed in 5 ml of solvents including ethanol (anhydrous, >99.5% Sigma-Aldrich), acetone (ACS reagent, >99.5% Sigma-Aldrich), cyclohexane (anhydrous, 99.5% Sigma-Aldrich), methanol (HPLC, >99.9% Sigma-Aldrich), acetaldehyde (ACS reagent, >99.5% Sigma-Aldrich), n-hexane (HPLC, >97% Sigma-Aldrich) for 2 hours.

Q6. What is atomic ratio from XPS? There might be some minor deconvolution issues (likely related to calibration), e.g., C-C (at ~283.2eV) and Ti-C seems to be off?

A6: Please find attached the atomic ratio of XPS and also the updated C 1s spectra after re-calibration.

Element	Atomic ratio (%)
Ti	39.2
C	27.3
O	19.4
F	14.1

Q7. Reference 10 does not seem to provide direct evidence on the claim that “volatile by-products including CO₂ and H₂O will interrupt the channel alignment and cause unwanted enlarged size distribution”. I would consider rewording this argument.

A7: We thank the reviewer for the constructive advice, and agree that the argument is based on our indirect observation from previous XRD results which often exhibit much larger FWHM after covalent channel modifications. Such disorder could stem from mixed reasons rather than just volatile CO₂ and H₂O. Therefore, we combine this argument with the other one and reword this section as below:

(Line 49 to 54) While providing reliable modification efficacy, covalent methods are largely discouraged from fulfilling their decoration potential due to the emergence of performance-compromising non-idealities to membrane structure in reaction process. These include unwanted defects, corrugations and interrupted channel alignment caused by harsh reaction conditions (e.g., high temperature and strong acid/base) and volatile by-products^{8,9,10} ...

Q8. AFM image is showing the MXene is somewhat oxidized. The rough and raised edges are usually an indication of oxidation. It will be better to replace this AFM image with a cleaner one.

A8: We apologize for the unsatisfactory AFM image quality and attribute the issue to the sample preparation using aging MXene suspension where nanosheets had initiated their oxidizing process. To improve the image quality, we produced a fresh batch of MXene nanosheets for AFM probing, as shown below and in updated Supplementary Fig. 1b. Compared with those in the previous figure, MXene nanosheets in the renewed one have clearly smoother edge without obvious oxidized dots. Meanwhile, the relatively rougher surface of the nanosheet as well as the mica support is another good implication of low oxidation state. This is because the oxidized parts normally have larger height and thus create greater contrast with the rest features of the image, making them look as flat as shown in previous AFM height profile. Therefore, the updated image better supports the claim of the prepared nanosheets having “low oxidation degree”. An inset is included in the figure to verify the homogeneous physicochemical property of MXene nanosheets taken from different samples.

Supplementary Fig. 1. Morphology characterizations of $\text{Ti}_3\text{C}_2\text{T}_x$ nanosheets, including (a) Transmission electron microscopy (TEM) image (inset, diffraction pattern) and (b) Atomic force microscopy (AFM) height profile (inset, nanosheets from a different sample).

...Meanwhile, the smooth nanosheet edges and the relatively low contrast between nanosheets and mica support indicated the rare presence of oxidized parts across the prepared nanosheets, thus proving their low oxidation degree.

Reviewer #3 (Remarks to the Author): The authors report on an interesting phenomenon where solvent molecules decorate the MXene surface to enhance sieving properties. Overall, it would be better if more results are provided to understand the sieving mechanism and the practicality of the method is further investigated. I believe this paper can be published after addressing some major revisions below:

Reply: We are greatly thankful for the reviewer's encouraging comments and suggestions to improve this work's overall quality. We have conducted additional experiments, characterizations and discussions where required.

Q1. Authors are encouraged to perform the same experiment and present data throughout the paper using another alcohol (such as methanol) to further prove the influence of hydrogen bonding. In fact, it is mentioned that methanol is used in the methods section, but no data can be seen.

A1: We followed the reviewer's advice to include methanol-treated membranes (MeOH-M) into the study and evaluated their characterization results and ion sieving performances, which are presented below and in updated Supplemental Figure 10. It is found that MeOH-M possessed similarly effective Na^+ sieving ability of $14.5 \text{ mmol} \cdot \text{m}^{-2} \cdot \text{hr}^{-1}$ and Na^+/H^+ selectivity of around 11. On top of this, in

combination with the other two reviewers' requests, we further treated MXene membranes with n-Hexane (Hex-M, non-polar) and acetaldehyde (MeCHO-M, polar yet aprotic) and found that they largely resembled the performance of their respective counterparts Cy-M and Ace-M that have had been discussed in the manuscript. We believe these outcomes not only help us confirm the influence of hydrogen bonding in the work, but also demonstrate the feasibility of channel decoration via non-covalent routes using a wide range of other solvents.

Supplementary Fig. 10. Na⁺ sieving and H⁺/Na⁺ selective performance of various membranes including Untreated-M, Hex-M, MeCHO-M and MeOH-M.

Additional MXene membranes were treated by methanol (MeOH), Acetaldehyde (MeCHO) and n-Hexane (Hex) and tested on their ion rejecting and selective performance. Similar to their counterparts, Hex-M, MeCHO-M and MeOH-M could improve Na⁺ sieving ability by 1.3, 3.9 and 25.3 times compared to Untreated-M, respectively. Accordingly, the H⁺/Na⁺ selectivity was increased from 1.8 to 1.9, 4.1 and 11, respectively.

Manuscript:

(Line 205-209) In addition, we also treated Ti₃C₂T_x membranes with a few more solvents including methanol (MeOH), Acetaldehyde (MeCHO) and n-Hexane (Hex), which respectively resemble EtOH, Ace and Cy in terms of chemical properties. Similar ion rejecting and selective performance trends were obtained, which demonstrated the reproducibility and universality of our method. (Supplemental Fig. 9)

Q2. How does the permeation rate compare in pressurized sieving systems instead of diffusion systems? It is doubtful whether the ethanol adsorption on MXene will remain intact if a stronger flow is applied to the system, as practical applications employ cross-flow filtration.

A2: Because of different driving force, water and ion flowing modes in pressurized sieving system (co-transport) and diffusion system (opposite-transport) are intrinsically varied, making a direct comparison on ion permeation rates very hard.

However, we do agree with the reviewer's concern that ethanol adsorption may not withstand much stronger disturbance in pressurized flows. To this end, we conducted another controlled experiment where EtOH-M underwent flowing water (1 Bar) in a dead-end cell for 20 mins, as shown above. As-treated membrane (EtOH-M-P) was then tested in diffusion cell on its Na⁺ permeation rate. It can be seen that EtOH-M-P shows larger ion permeation rate than EtOH-M, suggesting partial ethanol desorption from the membrane, as expected by the reviewer. This is indeed reasonable considering the possible impacts of flowing water on EtOH-M-P by (1) taking away the loosely bonded ethanol molecules; and (2) hydrating the membrane channels so as to expand channel size, which leads to weakened nanoconfinement effect as well as ethanol adsorption. On the other hand, however, we noticed the Na⁺ permeation in EtOH-M-P (95.4 mmol·m⁻²·hr⁻¹) is still slower than in Untreated-M (371.8 mmol·m⁻²·hr⁻¹) and similar to that in Ace-M (112.3 mmol·m⁻²·hr⁻¹). Combined with our DFT results in Fig S10c, this could imply that H-bond involving more -OH, such as HO-EtOH-OH, may remain stabilized under turbulent conditions.

While introducing more -OH presents a possible solution to more stable non-covalently decorated membranes used in future pressurized systems, we believe that the proposed membranes in their current form are already promising in other important applications such as electrolyzers and fuel cells, where no pressure is applied or required.

Q3. In the main text, it is suggested to briefly elaborate the physicochemical reason behind the blue shift of FTIR peaks after insertion of ethanol, instead of just allocating it to references

A3: We would like to first correct a typo in the previous description. The “blue shift” mentioned here should be “red shift” as a peak shift to lower wavenumber in FT-IR is generally referred as a red shift. Hydrogen bond-induced red shifts emerge because the H atoms attached to one electronegative atom has been attracted by another ($X-H\cdots Y$, or $O-H\cdots OH/O/F$ in our case), causing the electron cloud density change and thus elongation of participating bonds. As a result, the strength of these participating bonds will be reduced to show lower stretching vibration frequency (namely lower wavenumber). Accordingly, a brief explanation has been added in the manuscript and below:

(Line 88-91) This is largely because of the hydrogen bond formed between them and the -OH from the inserted ethanol, which causes the electron cloud density change and elongation of involving bonds. Consequently, the strength of the bonds will be decreased to show lower stretching vibration frequency in FTIR²⁰⁻²²

Q4. In figure 1b, why does the C-Ti-O and C-Ti-OH peak disappear after ethanol adsorption? As ethanol attaches non-covalently, the two peaks above should still remain on MXene.

A4: We apologize for the implicit figure presentation and caption that might have caused the reviewer’s confusion. In fact, Figure 1 a and b are respectively the Ti2p and C1s spectra of Untreated-M rather than EtOH-M. We displayed C1s to illustrate that C-O and C-F could be occasionally formed at the edge of $Ti_3C_2T_x$ nanosheets. However, to identify any possible changes on C-Ti-O and C-Ti-OH before and after ethanol adsorption is usually based on O1s spectra.

To address the concern of you and Reviewer #1, we added the XPS spectra of EtOH-M for a comprehensive comparison as shown in our response to Q7 of Reviewer #1. Please check for more information if necessary.

Q5. In figure 2d, the ‘onset’ point for the increase in permeation rate for untreated-M is around 96 hours of testing. It is curious whether this point never occurs for EtOH-M or whether this point arrives in extended testing times. Are the EtOH-M samples still stable for testing after more than 2 weeks?

A5: To this end, we conducted an extended long-term ion permeation test on EtOH-M and found its permeation rate remained almost unchanged until a slight increase on the 17th day, compared to a much earlier onset point for Untreated-M on the 4th day (Figure below). This indicates that EtOH decoration can maintain channel stability for at least 2 weeks, similar to one of the most durable membranes previously reported (*Nat Sustain* **3**, 296–302 (2020)). A minor change has been made in the manuscript from Line 122-124.

We also believe that with better storage of MXene dispersion and higher ethanol decoration efficiency, the stable operation time of EtOH-M can be further prolonged. Also, since this issue intrigued all the reviewers, we further verified and explained the ethanol-enhanced stability using a facile conductivity test in updated Fig. 2c and Supplementary Fig. 7. Please refer to Q3 of Reviewer #2 for more information.

Q6. Figure 4f in line 140 should be Figure 2f.

A6: We apologize for the typo, which has been corrected as it should be.

Q7. Please comment on the possibility of using this system in organic solvents toward OSN, etc.

A7: Based on the experimental and simulation results, we envisage that the proposed MXene system does have the potentials in certain solvent-solvent separation, gas separation and OSN processes where the desired species can form obviously stronger H-bond than the unwanted one, so the former can preferentially settle into the channel to further obstruct the transport of the latter, increasing separation efficiency. Possible separation settings can include C₂H₅OH/C₆H₅CH₃, H₂/N₂, and pharmaceutical separations. Moreover, this concept should be also applicable to other 1-nm or sub-1-nm systems that are able to form specific non-covalent interactions with species such as π - π interaction, though more works are needed to prove it. Accordingly, we added a related perspective to the conclusion:

(Line 280-288) This study proves the feasibility of non-covalent modifications in nanoconfined space, which is promisingly extendable to other appropriately sized nanochannels of various forms (e.g., porous and mixed matrix) and materials (GO, MoS₂, MOF, COF etc.). It also indicates that stable preferential settling of one species by non-covalent interactions can boost its selectivity against others in a separation process, providing a new perspective to understand transporting phenomena in 1-nm and Sub-1-nm channels. We believe these methodologies and underlying theories can be applied to develop and modulate membranes for a broader range of applications including organic solvent nanofiltrations, solvent separations, gas separations and beyond.

Q8. The mechanism of ion sieving seems to be highly influenced by the physical confinement induced by ethanol adsorption. How does the sieving performance of untreated MXene and EtOH-MXene turn out when these films are annealed at higher temperatures above the current 50 degrees? It is well known that the interlayer spacing significantly decreases when temperature above 120~150 degrees C

are used to anneal MXene films.

A8: We appreciate the reviewer’s thoughtful suggestion and carried extra XRD and ion-sieving test on membranes annealed at higher temperature (80°C and 120°C) after membrane preparation, the results of which are shown below. As it clearly demonstrates, annealing both membranes would cause the reduction of d -spacing and correspondingly decreased Na^+ permeation, primarily because of the “self-crosslinking” between neighboring nanosheets via their functional groups. (*ACS Nano* 2019, **13**, 10535–10544).

However, it was noteworthy that d -spacing reduction in EtOH-M was greater than in Untreated-M (e.g., 15.9 Å for Untreated-M and 15.5 Å for EtOH-M), while its Na^+ permeation rate change was less obvious. This led us to suppose that the formation of H-bond, due to its ability to weaken all participating chemical bonds (see your Q3), might result in the detachment of EtOH-decorated functional groups from MXene surface in high temperature, thus showing even smaller d -spacing. Meanwhile, the evaporation of EtOH molecules and other by-product during the thermal treatment could disrupt membrane layer alignment, making the annealed membrane’s performance inferior than expected. While these additional experiments provided interesting preliminary results worth studying, current data may not suffice to fully verify our hypothesis or to support us make any convincing claims. Therefore, we decide not to add this section in the updated manuscript, but only present it here for the reviewer’s information.

Response to Reviewers' Comments

Reference Number: NCOMMS-22-48007

Title: Non-covalently Decorated Ion-selective MXene Membranes Enabled by Nanoconfinement

Reviewer #1 (Remarks to the Author): This paper reports on a facile modification of Ti MXene with ethanol for membrane applications. The authors have characterized the membrane well and also proved the application perspective sufficiently. The title emphasize on nano confinement effect; however, it is not described well why it is happening and what is the advantage Etc. Please find my comments on this manuscript.

We are grateful for the reviewer's acknowledgement on our previous efforts, and helpful comments to enhance the manuscript quality. Please find below our further work to improve it following your advice.

Q1. Add reference to the "A typical example is aquatic-used graphene oxide (GO) and MXene channels that show effective width of ~0.8 nm" (Line 43-44)

A1: References have been added where necessary. Also, we made a slight change to the description from "~0.8 nm" to "~0.6 to 0.8 nm" to better reflect various real-world situations.

Q2. "Highly crystallized, monolayered Ti₃C₂T_x nanosheets (Supplementary Fig. 1) with low oxidation degree and small lateral size distribution (Supplementary Fig. 2) were etched from parent MAX phase". Low oxidation degree should be supported by TGA of the material under O₂ atmosphere. You can go through the reference - Liu, N., Li, Q., Wan, H. et al. High-temperature stability in air of Ti₃C₂T_x MXene-based composite with extracted bentonite. Nat Commun 13, 5551 (2022). <https://doi.org/10.1038/s41467-022-33280-2>.

A2: We greatly thank the reviewer for recommending a potential examining way on MXene oxidation. After careful reading the provided paper, we unfortunately realized that TGA might not be best way to identify the oxidation status of MXene dispersion mainly because of two reasons: (1) TGA is usually employed to characterize materials' thermal stability and thermal-related features such as anti-oxidation ability mentioned in this paper. However, it does not reveal the exact oxidation degree of the material before the test. This is in line with the concept of the provided reference, where the author conducted TGA to prove the improved anti-oxidation capability of MEB composites from pure MXene, rather than to compare their initial oxidation status; (2) The oxidation degree of MXene is more of a relative concept and so its precise determination via TGA, if ever possible, will rely on a comparison with an oxidation-free MXene nanosheets as reference. Unfortunately, MXene nanosheet synthesis via current methods is always accompanied by oxidation to a certain extent, which make a TGA-based analysis even less likely.

Instead, most studies so far resort to the combination of microscopic evidence to semi-quantitatively examine on the oxidation degree of freshly prepared MXene nanosheets, which involves direct MXene observations under TEM and SEM (*Chem. Mater.* 2017, 29, 4848–4856). As can be seen in updated Supplementary Fig. 1 and our response to Q1 of Reviewer #2, the TEM and AFM images showed even and smooth edge and clear single-crystal TEM diffraction pattern. These features are highly similar to the previously reported MXene nanosheets with low oxidation extent, thus supporting our claim. Please also be noted that although the recommended paper did not directly solve the oxidation issue, we found it very helpful in explaining the EtOH's protective role in improving EtOH-M stability and thus referenced it in the manuscript and Supplemental Fig. 7. Please refer to Q3 of Reviewer #2 for more information.

Q3. As described in Line 83, “-O, -OH and -F groups linked to Ti atoms, and C-F and C-O, presumably around the edge of nanosheets”. How did the authors come to the conclusion of edge functionalization than surface functionalization? Please explain. Also provide the elemental mapping of the edge functionalized MXene to further prove the statement.

A3: We apologize for the misleading sentence on nanosheet chemical composition. We followed the reviewer’s advice to add Energy-dispersive X-ray analysis (EDX) mapping to the $\text{Ti}_3\text{C}_2\text{T}_x$ nanosheet

under TEM, as shown below and in updated Supplementary Fig. 5. It can be seen that all the main elements including Ti, C, O and F (H is undetectable with current TEM EDX mapping precision) are evenly distributed and mixed, and most functional groups are located on the basal plane of the nanosheet. The original sentence in the manuscript is revised accordingly.

Supplementary Fig. 5:

Supplementary Fig. 5. X-ray photoelectron spectroscopy (XPS) spectra of MXene membranes including (a) O1s and (b) F1s. TEM image of MXene (c) and associated energy Dispersive X-ray (EDX) microanalysis.

...Energy Dispersive X-ray (EDX) microanalysis in selected area (c) show that the main composition elements of MXene including Ti, C, O and F are evenly distributed across its whole surface. This indicates that functional groups attached to the MXene are also well spread not only in the basal plane, but also along the edge of MXene nanosheets.

Manuscript:

(Line 82-85) In accordance with previous studies, X-ray photoelectron spectroscopy (XPS) and energy-dispersive X-ray (EDX) analysis revealed the membrane chemical composition, comprising -O, -OH and -F groups linked to Ti and C atoms across the basal plane and around the edge of nanosheets (Fig. 1 a, b, and Supplementary Fig. 5)^{15,16}

Q4. The concentration of MXene (mg/mL) used for the membrane preparation, volume of MXene solution taken, volume of ethanol or cyclohexane impregnated needs to accurately mentioned in the

description of Fig S4.

A4: We have elaborated the experimental section to include the above-mentioned details, and the updated version is as below and in the “Preparation of Pristine and Solvent Modified MXene Membranes” section. Please be noted that the “inconsistent sentences” as mentioned in Q19 remain here as these data have been added into the manuscript as requested by other reviewers.

(Line 301-314) Generally, after careful collection, MXene dispersion of around 0.7 mg/ml can be obtained... To fabricate laminar membranes, calculated amount of MXene would be filtrated on Nylon substrate (47mm diameter, 0.2 μm pore size) with vacuum assistance. For instance, to prepare a 300-nm thick membrane, dispersion containing 1.12 mg of MXene would be extracted and diluted by 200 ml deionized water. As-prepared membranes were dried in room temperature for 24 hr before further tests, treatments and characterizations. The solvent-modified membranes were prepared via a facile “immersion-evaporation” method, where these pristine membranes were immersed in 5 ml of solvents including ethanol (anhydrous, >99.5% Sigma-Aldrich), acetone (ACS reagent, >99.5% Sigma-Aldrich), cyclohexane (anhydrous, 99.5% Sigma-Aldrich), methanol (HPLC, >99.9% Sigma-Aldrich), acetaldehyde (ACS reagent, >99.5% Sigma-Aldrich), n-hexane (HPLC, >97% Sigma-Aldrich) for 2 hours.

Q5. SEM images of the membrane with and without solvent impregnation should be done to analyze the morphological changes if any.

A5: As suggested, we took and compared the digital photos, surface and cross-sectional SEM images of both Untreated-M and EtOH-M as shown below and in updated Supplementary Fig. 3. These photos reveal that (1) EtOH treatment does not cause any discernable visual difference across the membrane and its substrate; (2) Membrane surface morphology largely remain unchanged after EtOH treatment under SEM either in low resolution (typical ridge-and-valley structure) or high resolution (flatter surface); (3) EtOH treatment hardly alters the layered structure of the membrane. Therefore, it is reasonable to conclude that solvent impregnation can rarely results in obvious morphological changes.

Q6. Background correction of peak fitting should be included in XPS plots.

A6: Please see updated Figure 1 and Supplemental Fig. 5 for XPS plots with background correction.

Q7. The XPS provided is of synthesized MXene or ethanol treated MXene. Please specify. If not the latter, please provide the XPS of the MXene treated with ethanol to understand the change in surface functionalization of MXene.

A7: The XPS spectra provided in original manuscript is for Untreated-M. We followed the reviewer's suggestion to redo XPS for the comparison between Untreated-M and EtOH-M, which is shown as below. It can be seen that the introduction of ethanol leads to no emergence of new peaks and negligible shifts of current peaks (<0.1 eV) in Ti 2p, C 1s, O 1s and F1s spectra, thus denying the likelihood of obvious surface chemical modification caused by ethanol treatment.

The only changes that occur after ethanol treatment are the minor increase of C-O (C 1s, 286.1 eV) from 0.67% to 1.24%, and slight relative ratio rearrangement of O 1s species due to the introduction

of ethanol hydroxyl groups. This is reasonable considering the mild treatment conditions in our experimental protocol, where the drying temperature (50 °C) is way too low to initiate any chemical reactions or defunctionalisation. Please also be noted that the above data are only for comparison purposes here following the reviewer’s request, but not included in the updated manuscript. This is because even if we stick rigidly to the same MXene synthesis protocol, it is hard to produce materials with identical physicochemical properties every single time. Since all the theoretical analysis and calculations are based on previous XPS data, we decided to keep them in the manuscript to avoid data inconsistency.

Q8. Please provide the values of transmittance in FTIR and not in arbitrary units. It should be noted that the after-ethanol impregnation the peak corresponding to -OH should have higher intensity which is not observed in Fig 1c.

A8: We re-baselined the original spectra and created the two spectra with same Y-axis range to obtain an updated FTIR figure as below (left). As can be seen, EtOH-M does generate a -OH peak with increased intensity, broader peak area and shifted peak position, all indicating the formation of H-bond at this region. However, we will not be able to use the increased peak intensity of a proof of H-bond for two main reasons: (1) FTIR normally functions as a semi-quantitative characterization tool. Compared to minor changes to peak intensity, the emergence, vanish or shift of peaks conveys more accurate and convincing information from the sample; (2) In addition to EtOH adsorption, the -OH peak intensity can be also influenced by the local membrane thickness and functionality distribution of the tested area (a very small spot). Therefore, in the updated figures, we focused on the shift of MXene characteristic peaks and the appearance of -CH₃ and -CH₂-, which are intrinsic to EtOH.

Also, we provided relative transmittance instead of absolute transmittance value for updated Y-axis because the latter will cause a severe legibility issue to the figure. Please be noted this presentation method has been widely used in previous studies (*Nano Lett.* 2021, **21**, 3495–3502, *Nature Nanotechnology* **16**, 331–336 (2021)). Corresponding revisions have been made to updated Figure 1c

and in the manuscript in combination with Q3 of Reviewer #3, which is also shown as below.

(Line 85-93) These characteristic groups were also detected in Fourier-transform infrared spectroscopy (FTIR), but with clear red shift at some peaks after ethanol treatment (Fig. 1c). Two major peaks assigned to -OH (346 cm^{-1}), and Ti-O (626 cm^{-1}) moved to lower wavenumber of 3410 and 606 cm^{-1} ¹⁷⁻¹⁹. This is largely because of the hydrogen bond formed between them and the -OH from the inserted ethanol, which causes the electron cloud density change and elongation of involving bonds. Consequently, the strength of the bonds will be decreased to show lower stretching vibration frequency in FTIR²⁰⁻²². Also, two joint peaks emerging at 2920 and 2850 cm^{-1} , which are normally attributed to -CH₃/-CH₂-, also proves the successful insertion of EtOH molecules into MXene membranes...

Q9. An XRD pattern of treated and untreated MXene with scanning angle up to 70 degrees should be included to understand structural changes or oxidative changes as a result of prolonged heating as a part of membrane preparation.

A9: XRD scanning of both Untreated-M and EtOH-M in wider angle range ($4\text{-}70^\circ$) was conducted as per the reviewer's request, and the spectra were compared as below and in updated Supplementary Fig. 6. It can be seen that Untreated-M and EtOH-M show almost identical pattern in both dry and wet conditions, possessing an outstanding (002) peak at 6.5° and at 5.4° , respectively. This proves that 2 hr heating at 50°C did not cause any obvious structural or oxidative changes in EtOH-M. In fact, it is believed that 80°C (at vacuum) is the thermal threshold for any substantial change to happen (*ACS Nano* 2019, 13, 10535–10544).

hydration. Meanwhile, it is worth noting that, in both dry and hydrated conditions, the rest of XRD spectra (7° to 70°) for EtOH-M is also almost identical to that for Untreated-M. This implies that moderate heating of EtOH-M at 50°C will not cause any substantial structural or chemical changes to the membranes.

Q10. The oxidative stability of ethanol treated $\text{Ti}_3\text{C}_2\text{T}_x$ should be included as well (UV and TGA)

A10: The relative oxidative stability of EtOH-M to Untreated-M was proved by a commonly accepted method based on membrane conductivity test, and was explained in updated Supplemental Fig. 7. Please refer to Q3 of Reviewer #2 for detailed experimental and mechanism explanation.

Q11. In Fig 2b, the hydrated radii of Mg^{2+} is more than Ca^{2+} but permeability of Mg ions are higher than Ca. What could be the reason for this observation?

A11: Considering their error and the very close hydrated size of Mg^{2+} (8.6 Å) and Ca^{2+} (8.4 Å), we would say that the permeation rates of Mg^{2+} and Ca^{2+} are in the same range. Also, the permeation of ions in confined channels is dependent on a number of factors, including size, surface functionality. Hence, it is not a surprise that the permeation rate of Mg^{2+} can sometimes be slightly higher than that of Ca^{2+} . In fact, such phenomenon was also in very recent study of MXene-based membranes (*Nature Sustainability* **3**, 296–302 (2020)).

Q12. DMSO (polar and aprotic) also is proved to interact with MXene well. May be the studies on how it interacts with MXene should also be included.

A12: We appreciate the reviewer's suggestion in treating MXene membrane with another unique solvent and so we prepared DMSO-M following exactly the same experimental protocol for other solvents. Very sadly, our attempts to test its ion-sieving/selective performance were not successful due to the exceptional dissolving ability of pure DMSO on the substrate we used (we tried Nylon and PAN, Polyacrylonitrile). Over a short period of immersion, the top surface of nylon was dissolved, leading to the peeling off or disintegration of MXene membrane from the substrate and making further performance testing impossible.

While it is a shame that we could not directly investigate MXene-DMSO interaction, we instead chose acetaldehyde (MeCHO) to treat MXene membrane, and re-evaluated how these polar yet aprotic solvents interact with and decorate MXene channels. Combined with other reviewers' similar questions, detailed experimental data and discussion can be found in our response to Q1 of Reviewer #3.

Q13. Has the channel dimension been experimentally evaluated? If not, please include suitable references to prove the channel dimension to be less than 1nm which is requisite for ion separation.

A13: The most effective and universally acknowledged characterization tool to evaluate the channel dimension (or size) of 2D lamellar membranes is XRD, and our study followed the same test and

analysis protocol. Please refer to your Q20 to 22 where we provide detailed explanations to all your concerns with channel size.

Q14. How did the authors rule out the possibility of adsorption of positively charged ions through MXene that have high negative zeta potential value? Please clarify.

A14: It is true that adsorption of ions can happen in membrane-based operations due to electrostatic attraction. However, this process is commonly considered occurring during the very early stage of the test and reaching its equilibrium quickly.

To prove this viewpoint, we actively monitored Na^+ permeation through both 300-nm thick Untreated-M and EtOH-M in the first 5 mins using conductivity meter. Before exhibiting the typical “testing time-conductivity” linear relation, both Untreated-M and EtOH-M had a “dormant period” where the conductivity reading remained unchanged, a phenomenon largely attributed to ion adsorption. More importantly, such dormant period existed for less than 1 min. Considering that tests in our study could run up to 12 hours, it is reasonable to conclude that the impact of ion adsorption to the overall permeation rate is negligible.

Q15. Separation of solvent molecules requires a channel dimension of less than 2 nm (as stated in Line 32), why did the author choose separation of ions over solvent molecules?

A15: As the reviewer expected, the effective channel size of our Untreated-M is less than 0.8 nm (Please see Q20 for details) and this value falls into the range of solvent separation. However, we chose to focus on ion separation mainly because (1) this channel size is close to that of smallest salt ions (e.g., K^+ 0.66 nm and Na^+ 0.72 nm) and so a minor decoration to the channel can yield potentially high sieving and separating performance; (2) compared to hydrated ions who possess isotropic size, the size of solvent molecules can vary depending on specific molecular direction and this may make organic solvent less responsive to size-based separation mechanism; (3) as proved in this work, the transport of certain organic solvents (e.g., ethanol) can be largely affected by not only the channel size but also

its surface chemistry, potentially leading to unusual transporting phenomena.

Therefore, we started our research from aqueous system. However, we are also working on solvent-related separation on similar systems at the moment.

Q16. Fig. 4f shows that proton transport was also retarded in the ethanol anchored MXene channels, albeit by a much smaller extent compared to other salt transport. There is no Fig. 4F in the manuscript.

A16: We apologize for the typo here. This was meant to be Fig. 2f and it has been corrected to what it should be.

Q17. Abstract: Could the authors also mention impedance to 1-nm mass separations due to fouling?

A17: We definitely agree with the reviewer that fouling could be a factor that discourages 1-nm mass separations from being achieved. However, considering the main research focus of this work and more importantly, the abstract word limit (150 words) for the journal, we unfortunately are unable to introduce more fouling-related contents into the abstract, and we apologize for this.

Q18. Line 48- “The majority of alike modifications” can be changed to “A majority of such modifications”

A18: We have changed the vocabulary at the right spot as requested.

Q19. Inconsistencies in sentence construction- Introduction and materials and methods.

A19: We apologize for the inconsistencies showing up in these sections, and we believed that the reviewer was mainly referring to the “methanol-treated” and “hexane-treated” experiments that were missing from the previous manuscript. Please be noted these have been corrected in the revised manuscript with extra data added. Please see Q1 of Reviewer #3 for relevant information.

Q20. Could the authors please clarify what two-fold implication means in XRD? Perhaps, they mean a change in d spacing? Also, there is a slight change in FWHM, then how do the authors confirm if the hydrated interlayer channel size remained the same regardless of ethanol treatment? The slight variation in FWHM, could indicate a change in d spacing? Also, HRTEM images of the materials could be taken to understand if there are any corroborations with XRD.

A20: The two-fold implications obtained from XRD spectra were interlayer channel size and size distribution, respectively identified by peak position 2θ and FWHM. This is because 2D stacking membranes consist of crystalline 2D material layers arranged in a repeating pattern, so the distance between these repeating layers, namely d -spacing, is obtainable by applying Bragg's law $n\lambda = 2d \sin\theta$ at membrane stacking direction. Meanwhile, FWHM is a measure of crystallinity degree and thus is interpreted as the order of membrane stacking structure or size distribution. (e.g., *Nature* **550**, 380–

383 (2017) & *Nature Materials* **16**, 1198–1202 (2017)).

Our study followed the same protocol and calculated the *d*-spacing of hydrated Untreated-M channels as 16.5 Å from XRD (002) peak. Considering that *d*-spacing measures the “center-to-center” distance between two neighboring layers, the “effective” channel size should be calculated by subtracting $\text{Ti}_3\text{C}_2\text{T}_x$ thickness (around 8.8 Å) from the *d*-spacing (Figure above). This answers Q13 on how we identify Untreated-M channel size as 7.7 Å. On this basis, the almost unchanged (002) peak position and FWHM for EtOH-M proves two points. (1) The EtOH decoration does not cause a *d*-spacing change, showing that one EtOH molecule is only attached to one side of the channel rather than bridging both sides; (2) FWHM does not indicate *d*-spacing change, but *d*-spacing distribution. The slight change thus means that EtOH decoration posed very minor disruptions to the original stacking structure, which is an advantage of non-covalent decoration compared to disruptive covalent ones.

While HRTEM may present similar channel size information, it unfortunately does not work in our circumstance as the instrument cannot deal with water-containing membrane samples. Even if it can, the hydrated membranes will be dried in high TEM vacuum, thus presenting misleading information.

Q21. The authors mentioned an improvement in ion sieving efficiency, how can this be correlated to the XRD observation, where the authors state hydrated inter layer channel size remained the same.

A21: To help the reviewer understand in an easier way, we drew a schematic graph in Q20. Indeed, the XRD for Untreated-M and EtOH-M possess almost identical *d*-spacing. However, like we explained in Q20, this does not mean their effective channel size is the same. With ethanol decorated onto the channel wall, the steric hindrance will reduce the effective size of EtOH-M channels and thus lead to enhanced ion-sieving and ion-selective efficiency.

Q22. Also, could the authors estimate the reduction in channel width by studying the width before and after ethanol modification and also taking into account the size of ethanol molecule?

A22: As stated in Q21, the current model has already considered the reduced effective channel width due to ethanol molecules. However, to precisely quantify such reduction is very tough and tricky as the ethanol molecules at different spots can have varied folding and rotation degree, thus occupying the channel by different extent. For this reason, we only qualitatively explain the decoration effect in our manuscript, as highlighted below.

(Line 112-114) Once decorated with ethanol, it is expected that part of the channel is occupied to narrow down the effective channel width and obstruct the passage of ions.

Q23. Another point that is to be highlighted is the molecular shielding effect. The authors state that the permeation rate is unchanged in 168 hours. Is this performance dependent on flow rate? Also, this flow rate can be imagined from a point of view where ethanol completely covers the material surfaces. Given the already anionic nature of MXene before ethanolic coverage, do the authors anticipate a repulsion between modifying ethanol molecules and surface functionalities. Such a repulsion would lead to non-uniform coverage. (explain DFT in some details)

A23: In this study, membranes were tested in diffusion mode as shown in Fig 2a. The ion permeation rate of the Ethanol-treated membrane is unchanged in 168 hours under our current operation conditions. In diffusion mode, water and ion transport in opposite directions inside nanochannel. In principle, Ion permeation rate is impacted by water flow rate. Higher water flow rate often leads to lower ion permeation rate. However, this normally occurs only when the salt solution concentration is sufficiently enough to create enough osmotic pressure across the membranes (e.g., for NaCl the typical concentration is 1.0 mol/L). Considering that our solution concentration is 0.2 mol/L, the flow rate of water can be reasonably omitted.

DFT analysis (Figure 3d) shows that, more often than not, there are attractions between ethanol molecules and surface functionalities of MXene nanosheets, particularly between ethanol and -OH groups. More importantly, considering that MXene surface is normally highly functionalized due to its etching-based synthesis method, EtOH molecules are supposed to interact with more than one functionality during the modification process. On this basis, our further DFT results (Figure 4a) show that the interactions between modifying EtOH and functionalities are always attractive, though with different strength. Despite these attractions, we do agree with the reviewer that ethanol coverage is non-uniform. This is because the coverage is dependent on the functionality distribution on MXene surface, which is usually considered random and non-uniform.

Q24. In figure 7 the authors have taken 300 nm membrane for studies. Could you please explain why the 100 nm was not taken? Also, what is the influence of flow rate on S7 figure a? In addition, in the case of 300 nm thick membranes, it is mentioned that after 0.2 hours of ethanol treatment, further ethanol is not able to permeate the membrane, so in this case, what is the extent of modification of the membrane at 0.2 hours, is it 100% modification or is the modification % lower?

A24: The reason why 300 nm membranes were used in the testing is because its high mechanical strength. Moreover, we realized that 100-nm thick membranes could have uneven thickness and even defects across their surface. As stated in our response to Q23, higher water flow rate often leads to lower ion permeation rate in diffusion mode testing.

For the time-being, it is indeed very challenging to precisely quantify the extent of modification of the membrane at 2 hours, but the estimate on modification extent for saturated channels is possible.

According to our DFT results (Figure 4a), the modification extent should be theoretically 100% if we consider both physically and chemically adsorbed EtOH. However, if we only consider chemically stable EtOH modification, then the modification extent is 27% (Supplemental Figure 11). In an ideal situation, that is if EtOH can freely permeate throughout the membrane, these value should be the membrane modification extent.

Q25. The authors have not included a comparison with other similar literature reports (if any). May be a table to this end can answer this question.

A25: Please find updated Table S1 for a modification efficacy comparison of the proposed method to recently reported 2D channel modification methods, including covalent and physical ones. Such comparison is based on ion (Na^+ , K^+ and Li^+) sieving performance, and the performance enhancement after modification. Although this is not an exhaustive table, it still proves that our proof-of-concept non-covalent decoration can be as effective as other ways. Accordingly, the manuscript has been revised as below:

(Line 115-118) Such improved ion sieving performance is comparable to that of previously reported covalently- and physically-modified 2D membrane channels, suggesting similar decoration efficacy via non-covalent methods. (Supplemental Table 1).

Q26. In the permeation test, 0.2 M HCl was used, which is highly acidic. But the result for this is not included in Fig 2. Could the authors please include this as well.

A26: The H^+ permeation rate was actually included as the inset to Fig. 2f along with H^+/M^{n+} selectivity. However, we realized the font of the inset caption might be too small to attract attention. We apologize for it and have enlarged it to improve legibility. While to include this dataset into Fig. 2b is also a good option, we prefer to keep it at its original position because Fig. 2b is mainly devoted to exhibiting the superior sieving/rejecting ability of our membranes towards metal cations. Although H^+ transport is also moderately retarded after EtOH treatment, this limited sieving efficiency does not match the aim of Fig. 2b. Meanwhile, Fig. 2f is served as an outlook for the membrane's potential for future ion selective application, having H^+ permeation rate data presented here may be easier for our readers to grab the idea.

Q27. Also, the pH of the solution being passed may affect the affinity for ethanol toward the MXene membrane. Could the authors please shed some light into this aspect?

A27: Theoretically, the H-bond strength solely depends on the electronegativity of involved atoms or groups and the modification of pH and ionic strength (IS) itself will have very limited impact. However, in the current context, the extreme acidic environment is supposedly to enhance the H-bond strength in two possible ways: (1) excessive H^+ in the channel will largely discourage the deprotonation of -OH ($-\text{OH} = -\text{O}^- + \text{H}^+$) on MXene, allowing more -O-H to engage in H-bond establishment; (2) highly

acidic environment will decrease effective surface potential (Electro-attraction) of MXene and thus lead to narrower d -spacing. This will further reinforce nanoconfinement to boost H-bond strength. (I have actually finished a set of experiments to prove membrane's stability in acidic media, and this will put on here once the figure is ready).

Minor comments

Q1. Please italicize the letter 'x' in $Ti_3C_2T_x$

A1: Please be noted all the subscript letters "x" have been italicized as suggested.

Q2. A lot of emphasis is given to the phenomenon of 'nano-confinement'. Please elaborate the role of nano-confinement in the present context under the section of 'Ion sieving and separating performance of ethanol-treated membranes.

A2: In fact, the roles of nano-confinement is revealed and discussed in details in section 4 (Mechanisms underlying channel stability and ion-sieving ability) based on DFT result analysis. In general, its roles can be understood as: (1) limiting MXene-EtOH distance. The sub-1-nm MXene channel can limit MXene-EtOH distance to below 3.5 Å for much stronger H-bond, which ensures much more robust EtOH decoration than in bulk solution for improved and stable performance; (2) limiting EtOH's molecular orientation. This enables constant EtOH facing towards MXene surface with its OH, reducing its interaction with H₂O to further improve decoration stability.

Since section 4 is constructed to explain the experimental phenomena in section 2 (membrane performance), we accordingly changed the title of section 4 to "**Nanoconfinement-enabled channel stability and ion-sieving ability.**" to better serve its purpose.

Q3. JCPDS of MXene synthesized should be provided.

A3: While we tried our best to sort out the JCPDS of MXene in ICDD (International Centre for Diffraction Data), no such data could be found mostly because that MXene ($Ti_3C_2T_x$) can be synthesized using different methods, agents and formulas, each leading to potential unique XRD patterns and thus a standardized pattern is very hard to be defined. However, we can tell the XRD of our $Ti_3C_2T_x$, largely assembles to that of previous studies that adopted similar synthesis method and etching agents (*Nature Sustainability* **3**, 296–302 (2020)).

Q4. Novelty of the work should be included in the introductory and concluding sections.

A4: Please see the updated introduction (last paragraph) and conclusion where we have made according revision.

Q5. The efficiency of the membrane should well compare and validated with the already reported cation selective membranes based on MXene.

A5: Please see Major Q25 for relevant information, as we realized these two questions are highly similar. However, we mainly focus on cation-sieving membranes because (1) the testing parameters cation-sieving membranes are easier to compare; and (2) paper number of H^+/M^{n+} MXene membranes is currently low.

Q6. The membrane is selective to specifically cations and not all ions; appropriate title modification is suggested.

A6: We agree with the reviewer's suggestion and slightly modify the title to "Non-covalently Decorated **Ion-sieving** MXene Membranes Enabled by Nanoconfinement". This is because the manuscript is more about proposing and rationalizing a the nanoconfinement-induced membrane decoration strategy, which leads to very decent ion sieving performance. While H^+/M^{n+} selectivity is also tested, more efforts are needed for further performance enhancement and mechanism exploration.

Q7. In Supplementary fig. 13, why there is a decrease in the number of permeated K^+ at some points? What it means?

A7: The intermittent number decrease of K^+ is the synergic result of its Brownian motion in simulation process and our statistic analyzing method. To mimic real-world situation, all ions including K^+ were set free in the MD simulation box to allow a constant Brownian state when permeating MXene channels. Therefore, while these K^+ streamed into the channel under the chemical potential difference (generated by concentration gradient), a small proportion of them could indeed flow back at certain moments. Meanwhile, statistical analysis on ion permeation in MD followed an established method in previous works (Ref.) by counting the number of ions passing through the channel boundary per nanosecond. In our MD, such counting was conducted at a high frequency (40 times over 100 ns) to ensure data reliability. As a result, it is reasonable that ion backflow could be "occasionally" captured at some counting points to exhibit a dip observed in Fig. 13. Despite these occasional dips, the collected data still clearly show an overall increase over 100 ns and agree well with our experimental results.

Reviewer #2 (Remarks to the Author): This manuscript describes a simple method in regulating ion transport properties of Ti_3C_2Tx MXene membranes. The method is based on solvent treatment using protic solvent which "rearrange" the H-bond network within MXene membranes, providing a long-lasting effect in tuning the ion transport behavior. The manuscript also attempts to provide an explanation (including experimental data, DFT calculations and MD simulations) on the role of different solvent molecules in the nanochannel of the membrane. However, I believe additional analysis and discussion are needed on this front which will offer further insight to the ion transport mechanism. Nonetheless, I think this manuscript is well-written, and the results and arguments are

clearly articulated. I would recommend accepting the manuscript for publication after some minor revisions are made. My main comment regarding the manuscript is:

Reply: We greatly appreciate the positive and instructive comments from the reviewer, and devoted additional endeavors into addressing the reviewer’s questions in transport mechanism explanation and membrane characterizations.

Q1. The main argument of the work is that EtOH enhances the “nanoconfinement effect” within the MXene membrane. It is known that certain polar solvents have a strong affiliation (vs. water) to MXene, which is confirmed by the DFT calculation. While MD simulations reveal some information regarding the ion transport through the membrane. I think the underlying mechanism can be further articulated based on the data presented. To me the effect of EtOH seems to be two-fold: introducing steric effect/size exclusion, and also affecting the ion transport mechanism. The discussion on the latter is missing here.

Figure 2c shows that the activation energy increased considerably after EtOH treatment. It indicated that transport of Na⁺ ion is transitioning from Grotthuss-like (hopping) mechanism (20 kJ/mol) to a (sluggish) transport mechanism (39 kJ/mol). This is an evidence that EtOH in the channel is “breaking” the hopping pathway for Na⁺. I think providing some additional data (e.g. Arrhenius plot for proton and other selected cations) may help elucidate this better.

A1: We thank the reviewer for the insightful advice and we followed it to calculate the activation energy change of H⁺ transport from its Arrhenius plot before and after EtOH treatment, as shown below:

Indeed, as can be seen in H⁺ plot, H⁺ transport in Untreated-M channels were mostly undisturbed, requiring a fairly low activation energy E_a of 11.3 kJ/mol (0.12 eV). This well fell between 0.1 to 0.4 eV, the energy range commonly acknowledged for Grotthuss-like, or hopping, transport mode (*Anal. Chem.* 2014, **86**, 9362–9366). This is reasonable considering the appropriate effective Untreated-M channel size (7.7 Å to include two layers H₂O molecules) and its hydrophilic surface, which provides a continuous H-bond network across the majority of the channel for H⁺ to hop. In EtOH-M, however, higher E_a of 23.1 kJ/mol was observed, indicating H⁺ partial transition from hopping to sluggish

transporting mode. Combining with our DFT results (Fig 4a), we reckon this is because that the decorated EtOH molecules normally have their -OH facing toward the nanosheet and -C₂H₅ facing toward the channel. The replacement of original hydrophilic -OH and -O by more hydrophobic -C₂H₅ at these spots will break down the H-bond network and thus increase H⁺ transport resistance.

While the similar trend occurs to Na⁺ as well, we are not fully confident if the above transporting mode transition also applies. This is because whether metal ions like Na⁺ can “hop” is still in deep debate. However, it is obvious that the appearance of more hydrophobic in the channel -C₂H₅, besides posing steric effect, also causes extra barrier for Na⁺ dehydration process to further slow down its transport. The above data and mechanism analysis have been added in the reconstructed Figure 2 and corresponding paragraphs.

(Line 148-153) The enhanced selectivity was rationalized by Arrhenius plot of temperature against ion permeation rate (Fig. 2f). Although both ions experienced higher energy barrier in EtOH-M than in Untreated-M, the activation energy increase for Na⁺ (from 20.8 to 39.1 kJ mol⁻¹) is obviously more pronounced than that for H⁺ (11.3 to 23.1 kJ mol⁻¹). This implies that the decorated EtOH molecules created substantially larger barrier for Na⁺ to dehydrate while they only moderately interfered with the “hopping” (Grotthuss-like transport) of H⁺.

Additional minor comments are as follows:

Q2. Some section of the introduction has no citations (such as Line 41-51)

A2: A series of references have been added to relevant contents in these lines.

Q3. It would be useful to provide some information regarding the oxidation status of the membrane after test, as MXene is prone to oxidize in aqueous environment. EtOH treatment may be somewhat helpful to the oxidation stability.

A3: We were intrigued by the potential effect of EtOH in moderating MXene oxidation as much as our reviewers, and thus conducted a proof-of-concept experiment based on membrane conductivity (σ) testing, which is consider a reliable indicator of solid MXene oxidation degree (lower σ implies higher oxidation, *Matter* **1**, 513–526, August 7, 2019 & *npj 2D Mater Appl* **3**, 8 (2019)). The testing was carried out using a homemade assembly as shown below, where the Untreated-M or EtOH-M was firmly clamped and attached to conductive Cu foils for easy and stable measurement. The assembly was then kept in water for up to 10 days but taken out and thoroughly dried for resistance (R) measurement at the 1st, 3rd, 7th, and 10th day, and σ can be calculated by the equation below:

$$\sigma = \frac{L}{R \cdot A} \quad (L \text{ and } A \text{ respectively represents tested membrane length and cross-sectional area)}$$

Since L and A remained the same throughout the experiment, σ at different days could be simplified as $\sigma = 1/R$ and thus normalized for comparison purpose. It can be seen that both membranes experienced conductivity loss over the 10-day period, indicating ongoing MXene oxidation. However, such loss in Untreated-M was significantly faster, with conductivity plummeting to below 10% in 3 days, than in

EtOH-M whose conductivity could maintain above 50% of its initial value. Although a fully quantitative link can yet be established between conductivity loss and ion permeation increase, we believe that the above comparison well verifies the enhanced resistance of EtOH-M against oxidation.

We attributed the attenuated oxidation to the “shielding effect” of the absorbed EtOH molecules, which posed a steric hindrance to stop O_2 and H_2O from attacking MXene surface. Accordingly, the above methodology and reasoning are added as below and in updated Supplemental Fig. S7. It should be noted, however, that we did not directly use the membranes in ion-sieving performance testing for conductivity comparison. This was because that peeling these membranes from the device could cause minor damages to the membrane periphery and made the conductivity measurement inaccurate.

Supplementary Fig. 7. (a) Comparison of the anti-oxidation ability of Untreated-M and EtOH-M in aqueous conditions by (b) conductivity test. The normalized results are shown in (c).

Unwanted oxidation of MXene materials including $Ti_3C_2T_x$ is a universal problem in either dispersion or solid form (e.g., layered membranes). This happens when reactive H_2O and O_2 attack the Ti-C bonds of $Ti_3C_2T_x$ nanosheets, resulting in their de-functionalization and breaking. However, ethanol molecules attached onto the $Ti_3C_2T_x$ will prevent the direct contact of the material with reactants, which creates a “shield” to largely slow down the oxidation process (Fig. S6a). To prove this proposed mechanism, we designed a proof-of-concept experiment based on membrane conductivity (σ) testing, which is consider a reliable indicator of solid MXene oxidation degree (lower σ implies higher oxidation). The testing was carried out using a homemade assembly where

the Untreated-M or EtOH-M was firmly clamped and attached to conductive Cu foils for consistent measurement (Fig. S6b). Both assemblies were kept in water, taken out and thoroughly dried for resistance (R) measurement at the 1st, 3rd, 7th, and 10th day, and σ can be calculated by the equation below:

$$\sigma = \frac{L}{R \cdot A}$$

where L and A respectively represents tested membrane length and cross-sectional area, and further simplified as:

$$\sigma = \frac{1}{R}$$

, since L and A remained unchanged throughout the experiment. On this basis, σ measured at each testing point was normalized against its initial σ_0 and plotted in Fig. S6c. It can be seen that both membranes experienced conductivity loss over the 10-day period, indicating ongoing MXene oxidation. However, such loss in Untreated-M was significantly faster, with conductivity plummeting to below 10% in 3 days, than in EtOH-M whose conductivity could maintain above 50% of its initial value. Although a fully quantitative link can yet be established between conductivity loss and ion permeation increase, we believe that the above comparison well verifies the enhanced resistance of EtOH-M against oxidation.

Q4. Solvents such as methanol and n-hexane were also mentioned in the experimental section, but there seems to be no discussion in the main text. The required treatment time (Figure S7) was identified by using Na^+ ions. Would this change with other ion species? It will be good if results on other ions can tested to cross compare.

A4: We apologize for the inconsistency between the experimental section and main text, as the data for methanol and n-hexane treated membranes were collected to reassure the feasibility of the proposed method but not included in the previous manuscript. Encouraged by all the three reviewers, we brought these data back in the updated manuscript to help validate our methodology and theory in this work. Please refer to Q1 of Reviewer #3 for more detailed discussion and data.

In terms of the treatment time-dependent ion permeation tests, we followed the reviewer's suggestion and repeated the same experiment on K^+ and Li^+ whose data are shown as below and in updated Supplemental Figure 8. Meanwhile, the additional tests were applied to thickness-dependent ion permeation to ensure data consistency. It can be clearly seen that the permeation rate "plateau" shows for Na^+ are observed in K^+ and Li^+ cases as well.

Q5. The membrane drying time and ethanol treatment time described in the experimental section is confusing (Line 294-298) and does not seem to match with the date reported in Figure S7.

A5: We apologize for the confusing information here. We have revised the section as below.

(Line 309-314) The solvent-modified membranes were prepared via a facile “immersion-evaporation” method, where these pristine membranes were immersed in 5 ml of solvents including ethanol (anhydrous, >99.5% Sigma-Aldrich), acetone (ACS reagent, >99.5% Sigma-Aldrich), cyclohexane (anhydrous, 99.5% Sigma-Aldrich), methanol (HPLC, >99.9% Sigma-Aldrich), acetaldehyde (ACS reagent, >99.5% Sigma-Aldrich), n-hexane (HPLC, >97% Sigma-Aldrich) for 2 hours.

Q6. What is atomic ratio from XPS? There might be some minor deconvolution issues (likely related to calibration), e.g., C-C (at ~283.2eV) and Ti-C seems to be off?

A6: Please find attached the atomic ratio of XPS and also the updated C 1s spectra after re-calibration.

Element	Atomic ratio (%)
Ti	39.2
C	27.3
O	19.4
F	14.1

Q7. Reference 10 does not seem to provide direct evidence on the claim that “volatile by-products including CO₂ and H₂O will interrupt the channel alignment and cause unwanted enlarged size distribution”. I would consider rewording this argument.

A7: We thank the reviewer for the constructive advice, and agree that the argument is based on our indirect observation from previous XRD results which often exhibit much larger FWHM after covalent channel modifications. Such disorder could stem from mixed reasons rather than just volatile CO₂ and H₂O. Therefore, we combine this argument with the other one and reword this section as below:

(Line 49 to 54) While providing reliable modification efficacy, covalent methods are largely discouraged from fulfilling their decoration potential due to the emergence of performance-compromising non-idealities to membrane structure in reaction process. These include unwanted defects, corrugations and interrupted channel alignment caused by harsh reaction conditions (e.g., high temperature and strong acid/base) and volatile by-products^{8,9,10}...

Q8. AFM image is showing the MXene is somewhat oxidized. The rough and raised edges are usually an indication of oxidation. It will be better to replace this AFM image with a cleaner one.

A8: We apologize for the unsatisfactory AFM image quality and attribute the issue to the sample preparation using aging MXene suspension where nanosheets had initiated their oxidizing process. To improve the image quality, we produced a fresh batch of MXene nanosheets for AFM probing, as shown below and in updated Supplementary Fig. 1b. Compared with those in the previous figure, MXene nanosheets in the renewed one have clearly smoother edge without obvious oxidized dots. Meanwhile, the relatively rougher surface of the nanosheet as well as the mica support is another good implication of low oxidation state. This is because the oxidized parts normally have larger height and thus create greater contrast with the rest features of the image, making them look as flat as shown in previous AFM height profile. Therefore, the updated image better supports the claim of the prepared nanosheets having “low oxidation degree”. An inset is included in the figure to verify the homogeneous physicochemical property of MXene nanosheets taken from different samples.

Supplementary Fig. 1. Morphology characterizations of $\text{Ti}_3\text{C}_2\text{T}_x$ nanosheets, including (a) Transmission electron microscopy (TEM) image (inset, diffraction pattern) and (b) Atomic force microscopy (AFM) height profile (inset, nanosheets from a different sample).

...Meanwhile, the smooth nanosheet edges and the relatively low contrast between nanosheets and mica support indicated the rare presence of oxidized parts across the prepared nanosheets, thus proving their low oxidation degree.

Reviewer #3 (Remarks to the Author): The authors report on an interesting phenomenon where solvent molecules decorate the MXene surface to enhance sieving properties. Overall, it would be better if more results are provided to understand the sieving mechanism and the practicality of the method is further investigated. I believe this paper can be published after addressing some major revisions below:

Reply: We are greatly thankful for the reviewer's encouraging comments and suggestions to improve this work's overall quality. We have conducted additional experiments, characterizations and discussions where required.

Q1. Authors are encouraged to perform the same experiment and present data throughout the paper using another alcohol (such as methanol) to further prove the influence of hydrogen bonding. In fact, it is mentioned that methanol is used in the methods section, but no data can be seen.

A1: We followed the reviewer's advice to include methanol-treated membranes (MeOH-M) into the study and evaluated their characterization results and ion sieving performances, which are presented below and in updated Supplemental Figure 10. It is found that MeOH-M possessed similarly effective Na^+ sieving ability of $14.5 \text{ mmol} \cdot \text{m}^{-2} \cdot \text{hr}^{-1}$ and Na^+/H^+ selectivity of around 11. On top of this, in

combination with the other two reviewers' requests, we further treated MXene membranes with n-Hexane (Hex-M, non-polar) and acetaldehyde (MeCHO-M, polar yet aprotic) and found that they largely resembled the performance of their respective counterparts Cy-M and Ace-M that have had been discussed in the manuscript. We believe these outcomes not only help us confirm the influence of hydrogen bonding in the work, but also demonstrate the feasibility of channel decoration via non-covalent routes using a wide range of other solvents.

Supplementary Fig. 10. Na⁺ sieving and H⁺/Na⁺ selective performance of various membranes including Untreated-M, Hex-M, MeCHO-M and MeOH-M.

Additional MXene membranes were treated by methanol (MeOH), Acetaldehyde (MeCHO) and n-Hexane (Hex) and tested on their ion rejecting and selective performance. Similar to their counterparts, Hex-M, MeCHO-M and MeOH-M could improve Na⁺ sieving ability by 1.3, 3.9 and 25.3 times compared to Untreated-M, respectively. Accordingly, the H⁺/Na⁺ selectivity was increased from 1.8 to 1.9, 4.1 and 11, respectively.

Manuscript:

(Line 205-209) In addition, we also treated Ti₃C₂T_x membranes with a few more solvents including methanol (MeOH), Acetaldehyde (MeCHO) and n-Hexane (Hex), which respectively resemble EtOH, Ace and Cy in terms of chemical properties. Similar ion rejecting and selective performance trends were obtained, which demonstrated the reproducibility and universality of our method. (Supplemental Fig. 9)

Q2. How does the permeation rate compare in pressurized sieving systems instead of diffusion systems? It is doubtful whether the ethanol adsorption on MXene will remain intact if a stronger flow is applied to the system, as practical applications employ cross-flow filtration.

A2: Because of different driving force, water and ion flowing modes in pressurized sieving system (co-transport) and diffusion system (opposite-transport) are intrinsically varied, making a direct comparison on ion permeation rates very hard.

However, we do agree with the reviewer's concern that ethanol adsorption may not withstand much stronger disturbance in pressurized flows. To this end, we conducted another controlled experiment where EtOH-M underwent flowing water (1 Bar) in a dead-end cell for 20 mins, as shown above. As-treated membrane (EtOH-M-P) was then tested in diffusion cell on its Na⁺ permeation rate. It can be seen that EtOH-M-P shows larger ion permeation rate than EtOH-M, suggesting partial ethanol desorption from the membrane, as expected by the reviewer. This is indeed reasonable considering the possible impacts of flowing water on EtOH-M-P by (1) taking away the loosely bonded ethanol molecules; and (2) hydrating the membrane channels so as to expand channel size, which leads to weakened nanoconfinement effect as well as ethanol adsorption. On the other hand, however, we noticed the Na⁺ permeation in EtOH-M-P (95.4 mmol·m⁻²·hr⁻¹) is still slower than in Untreated-M (371.8 mmol·m⁻²·hr⁻¹) and similar to that in Ace-M (112.3 mmol·m⁻²·hr⁻¹). Combined with our DFT results in Fig S10c, this could imply that H-bond involving more -OH, such as HO-EtOH-OH, may remain stabilized under turbulent conditions.

While introducing more -OH presents a possible solution to more stable non-covalently decorated membranes used in future pressurized systems, we believe that the proposed membranes in their current form are already promising in other important applications such as electrolyzers and fuel cells, where no pressure is applied or required.

Q3. In the main text, it is suggested to briefly elaborate the physicochemical reason behind the blue shift of FTIR peaks after insertion of ethanol, instead of just allocating it to references

A3: We would like to first correct a typo in the previous description. The “blue shift” mentioned here should be “red shift” as a peak shift to lower wavenumber in FT-IR is generally referred as a red shift. Hydrogen bond-induced red shifts emerge because the H atoms attached to one electronegative atom has been attracted by another ($X-H\cdots Y$, or $O-H\cdots OH/O/F$ in our case), causing the electron cloud density change and thus elongation of participating bonds. As a result, the strength of these participating bonds will be reduced to show lower stretching vibration frequency (namely lower wavenumber). Accordingly, a brief explanation has been added in the manuscript and below:

(Line 88-91) This is largely because of the hydrogen bond formed between them and the -OH from the inserted ethanol, which causes the electron cloud density change and elongation of involving bonds. Consequently, the strength of the bonds will be decreased to show lower stretching vibration frequency in FTIR²⁰⁻²²

Q4. In figure 1b, why does the C-Ti-O and C-Ti-OH peak disappear after ethanol adsorption? As ethanol attaches non-covalently, the two peaks above should still remain on MXene.

A4: We apologize for the implicit figure presentation and caption that might have caused the reviewer’s confusion. In fact, Figure 1 a and b are respectively the Ti2p and C1s spectra of Untreated-M rather than EtOH-M. We displayed C1s to illustrate that C-O and C-F could be occasionally formed at the edge of $Ti_3C_2T_x$ nanosheets. However, to identify any possible changes on C-Ti-O and C-Ti-OH before and after ethanol adsorption is usually based on O1s spectra.

To address the concern of you and Reviewer #1, we added the XPS spectra of EtOH-M for a comprehensive comparison as shown in our response to Q7 of Reviewer #1. Please check for more information if necessary.

Q5. In figure 2d, the ‘onset’ point for the increase in permeation rate for untreated-M is around 96 hours of testing. It is curious whether this point never occurs for EtOH-M or whether this point arrives in extended testing times. Are the EtOH-M samples still stable for testing after more than 2 weeks?

A5: To this end, we conducted an extended long-term ion permeation test on EtOH-M and found its permeation rate remained almost unchanged until a slight increase on the 17th day, compared to a much earlier onset point for Untreated-M on the 4th day (Figure below). This indicates that EtOH decoration can maintain channel stability for at least 2 weeks, similar to one of the most durable membranes previously reported (*Nat Sustain* **3**, 296–302 (2020)). A minor change has been made in the manuscript from Line 122-124.

We also believe that with better storage of MXene dispersion and higher ethanol decoration efficiency, the stable operation time of EtOH-M can be further prolonged. Also, since this issue intrigued all the reviewers, we further verified and explained the ethanol-enhanced stability using a facile conductivity test in updated Fig. 2c and Supplementary Fig. 7. Please refer to Q3 of Reviewer #2 for more information.

Q6. Figure 4f in line 140 should be Figure 2f.

A6: We apologize for the typo, which has been corrected as it should be.

Q7. Please comment on the possibility of using this system in organic solvents toward OSN, etc.

A7: Based on the experimental and simulation results, we envisage that the proposed MXene system does have the potentials in certain solvent-solvent separation, gas separation and OSN processes where the desired species can form obviously stronger H-bond than the unwanted one, so the former can preferentially settle into the channel to further obstruct the transport of the latter, increasing separation efficiency. Possible separation settings can include C₂H₅OH/C₆H₅CH₃, H₂/N₂, and pharmaceutical separations. Moreover, this concept should be also applicable to other 1-nm or sub-1-nm systems that are able to form specific non-covalent interactions with species such as π - π interaction, though more works are needed to prove it. Accordingly, we added a related perspective to the conclusion:

(Line 280-288) This study proves the feasibility of non-covalent modifications in nanoconfined space, which is promisingly extendable to other appropriately sized nanochannels of various forms (e.g., porous and mixed matrix) and materials (GO, MoS₂, MOF, COF etc.). It also indicates that stable preferential settling of one species by non-covalent interactions can boost its selectivity against others in a separation process, providing a new perspective to understand transporting phenomena in 1-nm and Sub-1-nm channels. We believe these methodologies and underlying theories can be applied to develop and modulate membranes for a broader range of applications including organic solvent nanofiltrations, solvent separations, gas separations and beyond.

Q8. The mechanism of ion sieving seems to be highly influenced by the physical confinement induced by ethanol adsorption. How does the sieving performance of untreated MXene and EtOH-MXene turn out when these films are annealed at higher temperatures above the current 50 degrees? It is well known that the interlayer spacing significantly decreases when temperature above 120~150 degrees C

are used to anneal MXene films.

A8: We appreciate the reviewer’s thoughtful suggestion and carried extra XRD and ion-sieving test on membranes annealed at higher temperature (80°C and 120°C) after membrane preparation, the results of which are shown below. As it clearly demonstrates, annealing both membranes would cause the reduction of d -spacing and correspondingly decreased Na^+ permeation, primarily because of the “self-crosslinking” between neighboring nanosheets via their functional groups. (*ACS Nano* 2019, **13**, 10535–10544).

However, it was noteworthy that d -spacing reduction in EtOH-M was greater than in Untreated-M (e.g., 15.9 Å for Untreated-M and 15.5 Å for EtOH-M), while its Na^+ permeation rate change was less obvious. This led us to suppose that the formation of H-bond, due to its ability to weaken all participating chemical bonds (see your Q3), might result in the detachment of EtOH-decorated functional groups from MXene surface in high temperature, thus showing even smaller d -spacing. Meanwhile, the evaporation of EtOH molecules and other by-product during the thermal treatment could disrupt membrane layer alignment, making the annealed membrane’s performance inferior than expected. While these additional experiments provided interesting preliminary results worth studying, current data may not suffice to fully verify our hypothesis or to support us make any convincing claims. Therefore, we decide not to add this section in the updated manuscript, but only present it here for the reviewer’s information.

Reviewer comments, second round -

Reviewer #1 (Remarks to the Author):

Dear Authors,

I am extremely satisfied with the revised version. The authors have have addressed each and every points. Highly appreciated. I recommend the "accept" for manuscript in this present form.

Reviewer #2 (Remarks to the Author):

The authors have addressed all the questions and comments raised by the reviewer and have provided a range of additional data and discussion. I am happy for the manuscript to be accepted in its current form.

Reviewer #3 (Remarks to the Author):

The authors have well responded to the comments, and I believe that the paper can now be published.